# EffoVPR: Effective Foundation Model Utilization for Visual Place Recognition

**Issar Tzachor[1], Boaz Lerner[1], Matan Levy[2], Michael Green[1], Tal B Shalev[1], Gavriel Habib[1]**

**Dvir Samuel[1], Noam K Zailer[1], Or Shimshi[1], Nir Darshan[1], Rami Ben-Ari[1]**

[1]OriginAI, Israel [2]The Hebrew University of Jerusalem, Israel
issart@originai.co

## ABSTRACT

The task of Visual Place Recognition (VPR) is to predict the location of a query image from a database of geo-tagged images. Recent studies in VPR have highlighted the significant advantage of employing pre-trained foundation models like DINOv2 for the VPR task. However, these models are often deemed inadequate for VPR without further fine-tuning on VPR-specific data. In this paper, we present an effective approach to harness the potential of a foundation model for VPR. We show that features extracted from self-attention layers can act as a powerful re-ranker for VPR, even in a zero-shot setting. Our method not only outperforms previous zero-shot approaches but also introduces results competitive with several supervised methods. We then show that a single-stage approach utilizing internal ViT layers for pooling can produce global features that achieve state-of-the-art performance, with impressive feature compactness down to 128D. Moreover, integrating our local foundation features for re-ranking further widens this performance gap. Our method also demonstrates exceptional robustness and generalization, setting new state-of-the-art performance, while handling challenging conditions such as occlusion, day-night transitions, and seasonal variations.

## 1 INTRODUCTION

The task of Visual Place Recognition (VPR), also known as Geo-Localization, aims to predict the place where a photo was taken relying solely on the visual information in the image. This is typically done by an image retrieval approach (Arandjelovic et al., 2016; Hausler et al., 2021; Lu et al., 2024b; Berton et al., 2023; Lu et al., 2024a; Keetha et al., 2023) where a database of geo-tagged images is used, often referred as *gallery*. Real world data, including tagged VPR datasets, rely on two major sources for images: 1) car street-view and 2) people personal cameras (commonly mobile phones) (Arandjelovic et al., 2016; Torii et al., 2015; Warburg et al., 2020; Berton et al., 2022a; Barbarani et al., 2023). As a result, images contain natural objects that are irrelevant and sometimes misleading for VPR task. Figure 1 (top-row) shows an example, where people, vehicles, daylight, or camera angles might differ between images.

**Compact Features:** Modern models commonly use a deep neural network to extract a so-called global feature (a.k.a descriptor) for the query and gallery images (Levy et al., 2023; Jia et al., 2021; Radford et al., 2021; Li et al., 2022b; Liu et al., 2021). This so called *single-stage* approach is then followed by a nearest neighbor search in the feature space, to retrieve the matching candidates from the gallery. In practice, global features of the entire gallery need to be maintained in the RAM memory to enable fast retrieval. In large-scale and real-world scenarios, *reducing the memory footprint* for each image is crucial to ensure real-time applicability. Therefore, several works often promote compact features to achieve both accuracy and applicability (Zhu et al., 2023; Lu et al., 2024b; Berton et al., 2022a).

**Foundation Models:** Following best practices in computer vision, VPR methods are often initialized with ImageNet pre-trained weights (*e.g.* (Zhu et al., 2023; Berton et al., 2022b)), followed by finetuning on VPR datasets, *e.g.* MSLS (Warburg et al., 2020), or trained from scratch (Wang et al., 2022; Hausler et al., 2021; Arandjelovic et al., 2016). Recent advancements in VPR exploit the capabilities of foundation models (Keetha et al., 2023; Izquierdo & Civera, 2024; Lu et al., 2024b;a)

such as DINOv2 (Oquab et al., 2023), a transformer (Vaswani et al., 2017) based model trained by self-supervised learning. Recent approaches (Lu et al., 2024b;a; Izquierdo & Civera, 2024) argue that using vanilla DINOv2 (without fine-tuning) is ineffective. They criticize this approach for failing by capturing dynamic and irrelevant elements (pedestrians or vehicles). Some methods propose to address this issue by integrating learned adaptors into the foundation model architecture (Lu et al., 2024b;a) or changing the standard fine-tuning scheme with a unique pooling mechanism (Izquierdo & Civera, 2024), in order to facilitate the foundation model adaptation to the VPR task. In contrast, Anyloc (Keetha et al., 2023) suggested to use DINOv2 as a general-purpose feature representation at *zero-shot* for various localization tasks.

**Pooling Methods:** Typically, VPR methods (Izquierdo & Civera, 2024; Lu et al., 2024b;a; Berton et al., 2023; Zhu et al., 2023; Berton et al., 2022a) adhere to a conventional approach of initially extracting local features from a pretrained model, then using pooling methods such as GeM (Radenović et al., 2018) or NetVLAD (Arandjelovic et al., 2016), to obtain a global feature. Methods using VLAD (Jégou et al., 2010), such as AnyLoc (Keetha et al., 2023), or NetVLAD (Arandjelovic et al., 2016) for aggregating local features necessitate learning a dictionary for each specific gallery. These models tend to "overfit" to the particular gallery distribution, which hampers their generalization. Additionally, they often require fine-grained clustering, leading to high-dimensional feature vectors. Furthermore Shao et al. (2023), highlighted GeM shortcomings, as hyper-parameter sensitivity, spatial information loss, and convergence complications.

In this paper, we challenge the necessity of the external pooling layers, and suggest to utilize the internal ViT self-attention mechanism for an implicit pooling (Li et al., 2022a). Apparently, a straightforward approach that trains with the class $[CLS]$ token solely (instead of finetuning by all patch tokens) implicitly aggregates local features into a global representation. This approach removes the reliance on external specialized components or learned features, which often result in excessively large feature representations (Izquierdo & Civera, 2024; Lu et al., 2024a;b; Ali-bey et al., 2024). Moreover, this approach facilitates simple dimensionality reduction, allowing to achieve state-of-the-art results even with compact feature representations (see Figure 1 - bottom row).

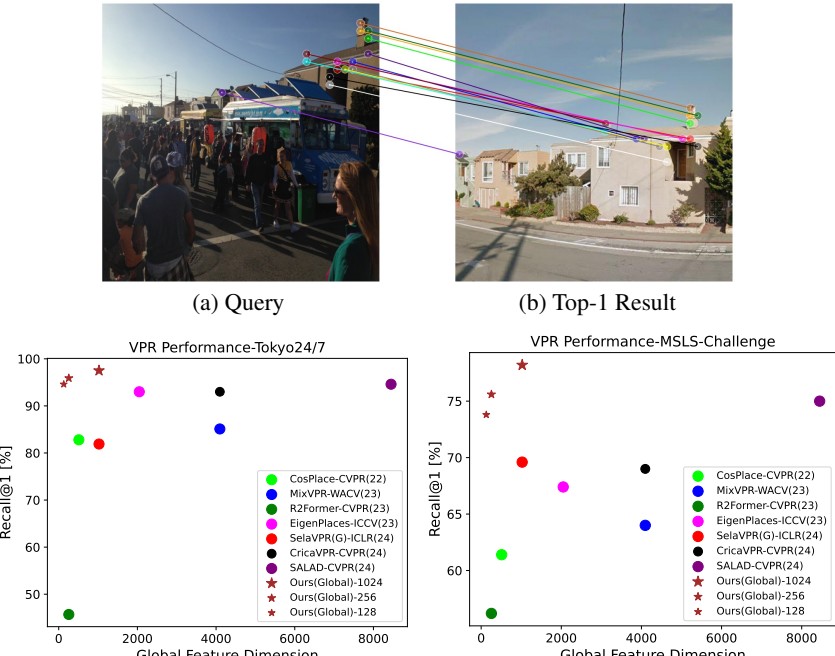

Figure 1: EffoVPR showcase. **Top row:** we show a challenging query image (a) and EffoVPR's top-1 candidate (b), retrieved from a gallery of 2.8M geo-tagged images of SF-Occ. EffoVPR demonstrates high capability in handling transitional objects and large obstructions. **Bottom row:** We present Recall@1 performance of EffoVPR using global features against feature dimensionality. While the current leading methods achieve their performance using large features, EffoVPR demonstrates top performance even with an extremely compact feature size.

**Re-ranking:** A popular strategy to improve the accuracy is to conduct a *two-stage* search, with subsequent similarity search on the top-k ranked results. This step is commonly performed by matching local key-points and their corresponding descriptors. We build on prior work highlighting the robust properties of SSL-trained models, *e.g.* their semantic and instance-level capabilities Oquab et al. (2023); Caron et al. (2021); Park et al. (2023); Keetha et al. (2023), with Amir et al. (2022), exploring the roles of self-attention facets across various tasks. In this paper, we propose a matching method for VPR that uses specific internal ViT features, effective even in a zero-shot setting, outperforming current zero-shot methods and competitive with several fine-tuned VPR approaches. Fine-tuning the DINOv2 backbone for the VPR task further enhances the quality of the features and the overall performance of our matching method, achieving state-of-the-art results.

We hereby present an **Ef**fective **fo**undation based VPR method called EffoVPR that achieves superior single-stage performance without requiring additional components. Unlike more intricate DINOv2-based methods (Lu et al., 2024a) that rely on adapter components (Lu et al., 2024b) or special aggregation techniques (Izquierdo & Civera, 2024), EffoVPR delivers state-of-the-art results across multiple datasets while maintaining a compact feature representation. Additionally, EffoVPR demonstrates strong generalization and robustness across diverse cities and landscapes, effectively handling challenges such as occlusions, time differences and seasonal changes as demonstrated in Figure 1, which illustrates significant occlusion.

To summarize, our contributions in this work include:

1. We introduce a novel image matching method for VPR that relies on keypoint detection and descriptor extraction from the intermediate layers of a foundation model. This method outperforms previous zero-shot approaches while showing comparable results with trained VPR methods on a few datasets.

2. We challenge the necessity of aggregation methods commonly used in various state-of-the-art methods, and propose a simple yet effective alternative that significantly improves performance and feature compactness.

3. We propose an effective approach that involves fine-tuning, followed by re-ranking the top candidates using our matching method. This approach significantly boosts performance, even with a very small number of candidates.

4. We achieve state-of-the-art results on numerous benchmarks, demonstrating outstanding robustness across varying scenes *e.g.* day vs. night, different seasons and occlusions, while achieving feature compactness that is often orders of magnitude better than prior methods.

## 2    RELATED WORK

Traditional approaches utilized a single-stage approach for Visual Place Recognition (VPR), using SIFT (Lowe, 2004) SURF (Bay et al., 2006), or RootSIFT (Jin et al., 2021), focused on matching queries to gallery images through the use of image local feature matching. Two-stage methods (Hausler et al., 2021; Wang et al., 2022; Zhu et al., 2023; Lu et al., 2024b) entail an initial ranking, based on global representation similarity, succeeded by re-ranking the top-K retrieved candidates in the second stage, utilizing local features.

First deep learning approaches for the VPR task used CNN (Arandjelovic et al., 2016; Radenović et al., 2018; Jin Kim et al., 2017) that was later replaced with Vision Transformers (ViT) (Wang et al., 2022; Zhu et al., 2023; Keetha et al., 2023; Lu et al., 2024b;a; Izquierdo & Civera, 2024). TransVPR (Wang et al., 2022) and R2Former (Zhu et al., 2023) suggested the application of transformers for VPR and adopted a two-stage approach that included re-ranking. However, training from scratch or initialized on ImageNet, and lack of effective view-variability in training has restricted their performance as they introduce a significant challenge in selecting appropriate positive and hard negative images (Arandjelovic et al., 2016; Warburg et al., 2020; Zhu et al., 2023; Berton et al., 2022a). EigenPlaces (Berton et al., 2023) introduced a novel training paradigm that organizes the training data into classes, with each class containing multiple viewpoints of the same scene. By employing a classification loss rather than the conventional contrastive loss (used in (Lu et al., 2024b; Izquierdo & Civera, 2024; Lu et al., 2024a)), the resulting model demonstrated high resilience to varying viewpoints in testing. Hence, we adopt a similar strategy for our training process.

A handful of visual foundation models (VFMs) have recently emerged as the backbones for numerous tasks. VFMs like CLIP (Radford et al., 2021), DINOv2 (Oquab et al., 2023), SAM (Kirillov et al., 2023) use ViT that are trained with distinct objectives, exhibiting unique characteristics for various downstream applications. In VPR, SALAD (Izquierdo & Civera, 2024) proposed fine-tuning the pre-trained DINOv2 model to improve its performance for VPR task. Their single-stage approach involves pooling local features from the output layer of DINOv2 replacing NetVLAD's soft-assignment to clusters by an optimal transport methodology. Notably, their most significant results are achieved with large features exceeding 8K dimension, which adversely impacts memory consumption. Using DINOv2 as their backbone, SelaVPR (Lu et al., 2024b) and CricaVPR (Lu et al., 2024a) took a different path and avoided fine-tuning of the model, proposing the incorporation of a trainable adapters integrated into DINOv2 architecture. SelaVPR built above DINOv2's output tokens, while discarding the standard [CLS] token. Their re-ranking strategy does not require spatial verification and is based on a newly learned patch level (local) features. CricaVPR instead, suggests learning a features pyramid above DINOv2's output tokens, to learn a special viewpoint robust encoder. Both, SelaVPR and CricaVPR incorporate adapters in the DINOv2 architecture, using GeM pooling for feature aggregation, trained with the common contrastive loss.

Current VPR methods predominantly, derive a global representation by aggregating local features obtained from the *output layer* of a trained backbone. This aggregation is often conducted using *external* modules for learnable pooling techniques, such as VLAD (Vector of Locally Aggregated Descriptors) (Jégou et al., 2010), SPoC (Sum-Pooled Convolutional features) (Babenko & Lempitsky, 2015), RMAC (Regional Maximum Activation of Convolutions) (Tolias et al., 2015), the known NetVLAD (Arandjelovic et al., 2016) or the nowadays widely-used GeM (Generalized Mean) pooling layer (Radenović et al., 2018; Shao et al., 2023), as utilized in various studies (Berton et al., 2021; Garg & Milford, 2021; Ge et al., 2020; Warburg et al., 2020; Hausler et al., 2021; Izquierdo & Civera, 2024; Lu et al., 2024b;a). However, NetVLAD is encumbered by high computational costs and dimensional complexities (with $\sim 32K$ dimensional feature) (Berton et al., 2022a), while GeM encounter convergence issues and require hyperparameter tuning (Shao et al., 2023). Recently, more sophisticated approaches were suggested. SALAD (Izquierdo & Civera, 2024) applied optimal transport for local features aggregation, while BoQ (Ali-bey et al., 2024) aggregated features by learning a Bag-of-Queries along cross-attention. In this paper, we propose a method that, unlike previous approaches, utilizes the existing internal aggregation layer in DINOv2 without the need for additional pooling components or adaptors, achieving SoTA results with highly compact features.

DINOv2 deep features have been utilized for various tasks, *e.g.* part co-segmentation (Amir et al., 2022), establishing semantic correspondences between image pairs (Zhang et al., 2023; Shtedritski et al., 2023; Zhang et al., 2024) or visual appearance transfer (Tumanyan et al., 2022), using internal features. Motivated by these studies we present a novel local matching method for VPR that detects keypoints and extracts their corresponding descriptors using a unique combination of ViT features.

Foundation models has shown strong applications even at zero-shot setting (without requiring any fine-tuning). In VPR, AnyLoc (Keetha et al., 2023) proposed a single-stage solution that leveraged a DINOv2 pre-trained model alongside dense local (patch-level) features. By employing VLAD for feature pooling across multiple layers, AnyLoc approach produces an extremely large global representation of $\sim 49K$ dimensions, subsequently reduced to 512D through PCA whitening, but in expense of lower performance. AnyLoc's VLAD aggregation tends to fail in generalizing to queries with large time-gaps, day vs. night or of different season. In this paper, we propose a two-stage *zero-shot* approach for VPR that first performs global ranking, followed by re-ranking. Moreover, our approach is adaptable to fine-tuned models. Remarkably, after fine-tuning, it attains state-of-the-art results while preserving compact features, which is essential given practical memory constraints.

## 3 METHOD

We start by outlining our model training strategy, followed by our local feature extraction method. Our approach stems from the insight that DINOv2 foundation model encapsulates robust features suitable for VPR tasks (Keetha et al., 2023). The core idea is to employ specific facets within the self-attention layer as reliable keypoint detectors and others as robust descriptors, for re-ranking.

We explore ViT (Dosovitskiy et al., 2021) feature maps as local patch descriptors. In a ViT architecture, an image is split into $p$ non-overlapping patches which are processed into tokens by linearly

projecting each patch to a $d$-dimensional space, and adding learned positional embeddings. An additional [CLS] token is inserted to capture global image properties. The set of tokens are then passed through $L$ transformer encoder layers, each consists of normalization layers, Multihead Self-Attention modules, and MLP blocks. In ViT, each patch is directly associated with a set of features, a *Key*, *Query*, *Value* that can be used as patch descriptors. We utilize the self-attention matrices at layer $l$, as $\{K, Q, V\} \in \mathbb{R}^{(p+1) \times d}$, with $p$ indicating the number of patches resulting $p + 1$ tokens (number of patches in the image plus one added global token) and $d$ standing for the embedding dimension. The self-attention function at layer $l$ is then given by:

$$\text{Attention}(Q, K, V) = \text{Softmax}(\frac{K^T Q}{\sqrt{d}})V \qquad (1)$$

Note that for sake of brevity we omit the index $l$. We indicate the key feature of the $[CLS]$ token at layer $l$ by $k_{cls} \in K$. Each image patch therefore is directly associated with the set of features $\{q_i, k_i, v_i\}_{i \in [p+1]}$ including its query, key and value at certain layer $l \in [1, L]$, respectively.

In the global search stage, the aim is to perform an efficient search across a vast corpus of images in a gallery. To this end, a shared embedding space for query and gallery images is typically utilized to identify the most similar images and rank them accordingly. In the re-ranking stage, the top-K nearest neighbors (candidates) are further refined by evaluating the mutual similarity between their local features and those of the query image. Throughout both stages, we use a ViT (Dosovitskiy et al., 2021) encoder as our backbone model. Our EffoVPR retrieval method is presented in Figure 2.

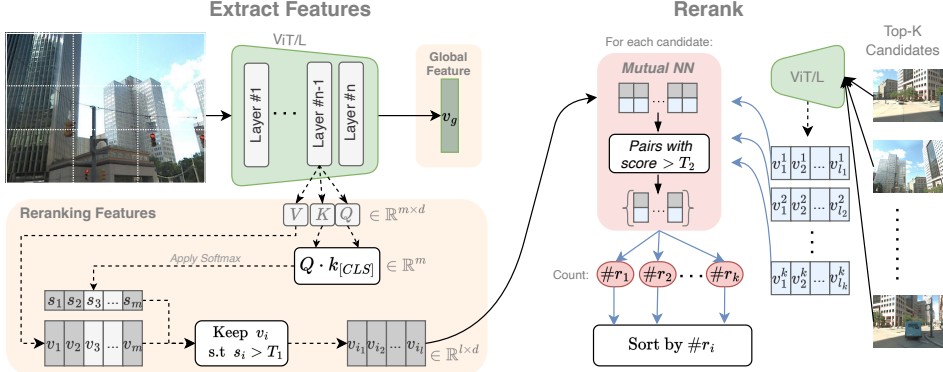

Figure 2: An overview of EffoVPR. **Left:** During inference, we identify the nearest neighbors of the query by using the [CLS] token as the global representation for each image ($v_g$). For the second re-ranking stage, we extract (dashed-line) intermediate features, and utilize $S$, the partial attention map, for *keypoint selection* (with a predefined threshold $T_1$), while employing the Value facet $V$ as the corresponding *keypoints descriptors*. **Right:** Lastly, we re-rank the top-K candidates from the first stage based on the count of strongly connected mutual nearest neighbors (MNN) with a score exceeding a predefined threshold (denoted as $T_2$).

## 3.1 TRAINING STRATEGY

Our training strategy employs a classification loss applied on the output $[CLS]$ token. Following EigenPlaces (Berton et al., 2023), we partition the map into cells measuring $15 \times 15$ meters each, where cells define classes. Each class contains images capturing the same location from various viewpoints, enhancing the model's ability to accurately recognize locations despite substantial changes in the captured view angle. To enforce class discrimination while ensuring viewpoint invariance, we utilize Large Margin Cosine Loss (CosFace) (Berton et al., 2022a; Wang et al., 2018).

We initialize our ViT (Dosovitskiy et al., 2021) backbone with pre-trained DINOv2 weights (Oquab et al., 2023; Darcet et al., 2023). To retain the rich visual representations learned during pre-training, while adapting the model for the VPR task, we fine-tune only the final layers of our backbone. Note that due to the inherent design of the ViT architecture, which incorporates a self-attention mechanism (Eq. (1)), the global feature is implicitly trained to aggregate local features without the

need of additional specialized components. For learning more compact global feature we simply add a linear layer on top of the output [CLS] token.

## 3.2 INFERENCE STRATEGY

**Global ranking stage:** Post-training, we extract the global feature of an image from the $[CLS]$ output token of the penultimate classification layer. We then retrieve the K nearest neighbors of the query's global feature.

**Re-ranking stage:** We start by extracting patch local features from each candidate among the top-k retrieved images in the previous global stage, and re-rank the candidates based on these features. Local features are derived from the intermediate layer $l$ of ViT, utilizing the self-attention matrices, thus their computation is integrated into the process of computing global features and does not necessitate recalculation. For *keypoints descriptors*, we find that $V_l$ matrix in Eq. (1) most effectively captures the "instance" properties of the keypoints, making it the most promising facet for the task at hand (see analysis in Tab. S3 in Appendix). Then, for *keypoint selection*, we leverage the model's internal prioritization, to identify discriminative features and extract the attention map: $S := \text{Softmax}(Q_l \cdot k_{cls}) \in \mathbb{R}^p$, which represents the attention of each Value feature with the global $[CLS]$ image representation key-token $k_{cls}$. We therefore select a subset $\mathcal{V}$ of the values $\mathcal{V} \subseteq \{v_1, ..., v_p\} := V_l$ as the image's local features, based on the score $S$: $\mathcal{V} := \{v_i \mid S_i > T_1\}$ for a predefined threshold $T_1$. Note that the number of selected local features may vary between images.

Next, given the query's local features $\mathcal{V}$ and a candidate's $\mathcal{V}'$, we calculate the pairs of mutual nearest neighbors (MNN) between $\mathcal{V}$ and $\mathcal{V}'$ by the number of feature pairs that are the nearest neighbor of each other. Our observation revealed that applying a threshold to the MNN scores enhances the model's resilience to clutter and directs its attention to the pertinent key-points for accurate matching. We therefore count only the pairs with cos-similarity higher than a predefined threshold $T_2$. We formulate that process in Equation (2):

$$\text{MNN}(\mathcal{V}, \mathcal{V}') := \{(v_i, v'_j) \in \mathcal{V} \times \mathcal{V}' \mid v_i := NN(V, v'_j) \text{ and } v'_j := NN(V', v_i), \ v_i^T v'_j > T_2\} \quad (2)$$

Finally, the re-ranking stage concludes by sorting the top-K candidates based on their MNN counts. Note that our re-ranking strategy, which matches local features, does not require any additional learning, optimization, or spatial verification. The thresholds are established once and remain fixed across all test sets (20 different scenarios).

**Zero-shot:** For the first stage ranking we use the [CLS] token from vanilla-DINOv2. The results are then refined by re-ranking, while employing our suggested features $\mathcal{V}$ with MNN in Eq. (2).

Our design parameters include the two key values of $T_1, T_2$. Threshold $T_1$ is used to select strong keypoints for image matching, typically focusing on more significant areas. The threshold $T_2$ facilitates the selection of strong matching pairs, filtering out weak correspondences that could result in false image matches. For a detailed ablation study on these thresholds refer to Appendix B.

## 4 EVALUATION

In this section, we compare our approach with several SoTA VPR methods following the common VPR Benchmarks (Berton et al., 2022c). We propose, a single-stage (EffoVPR-G) and two-stage, that includes a re-ranker (EffoVPR-R) approaches with backbone trained on the publicly available SF-XL (Berton et al., 2022a) dataset containing streetview of San Francisco. For more implementation details see Appendix. We then test EffoVPR on a large number of diverse datasets (20), including *e.g.* Pitts30k (Arandjelovic et al., 2016), Tokyo24/7 (Torii et al., 2015), MSLS-val/challenge (Warburg et al., 2020) Nordland (Sünderhauf et al., 2013) and more, exhibiting a wide variety of conditions, including different cities, day/night images, and seasonal changes. Note that MSLS Challenge (Warburg et al., 2020) is a hold-out set whose labels are not released, but researchers submit the predictions to the challenge server. Further details on the benchmarks are in the Appendix.

Datasets with gallery made from street-view images and with largest viewpoint variance, include Tokyo 24/7 (Torii et al., 2015) and SF-XL (Wang et al., 2018; Barbarani et al., 2023), where the query images are collected from a smartphone, usually from sidewalks. Most datasets are from urban footage, with the main exception being Nordland (Sünderhauf et al., 2013), which is a collection of

photos taken across different seasons with a camera mounted on a train. Some datasets present various degrees of day-to-night changes, namely MSLS (Warburg et al., 2020), Tokyo 24/7 (Torii et al., 2015), SF-XL (Berton et al., 2022a) SVOX-Night (Berton et al., 2021). AmsterTime (Yildiz et al., 2022) contains grayscale historical queries and modern-time RGB gallery images, making it the only dataset with large-scale time variations of multiple decades.

We follow common evaluation metric used in previous works *e.g.* (Berton et al., 2022c; 2023; Zhu et al., 2023; Lu et al., 2024b;a; Arandjelovic et al., 2016) and use 25 meters radius as the threshold for correct localization and report Recall@K metrics for K=1,5,10. For Nordland we evaluate $\pm 10$ frames as the common evaluation protocol used in (Berton et al., 2022c; Hausler et al., 2021). For a more comprehensive description of all 20 datasets and implementation details see Appendix.

## 4.1 Zero-shot performance

We present the performance of our zero-shot method (EffoVPR-ZS) in Table 1 compared to two zero-shot alternatives, the recently published AnyLoc, and DINOv2 global feature (using the output [CLS] token), without finetuning, where EffoVPR-ZS re-ranks its top-100 global retrieved candidates. The results show that our method significantly improves over the baseline and is

Table 1: Comparison on Zero-Shot with R@1.

|  | Pitts30k | Tokyo24/7 | MSLS-Val | Nordland |
|---|---|---|---|---|
| DINOv2 | 78.1 | 62.2 | 47.7 | 33.0 |
| Anyloc | 87.7 | 60.6 | 68.7 | 16.1 |
| EffoVPR-ZS | **89.4** | **90.8** | **70.3** | **57.9** |

superior to AnyLoc. Note the significant gap for more challenging scenarios of Tokyo24/7 and Nordland exhibiting day vs. night and seasonal variations. AnyLoc tends to fail in these challenging scenarios as its VLAD aggregation learned (in unsupervised manner) on the gallery can not generalize well to challenging out-of-distribution queries. In Figure 3a we compare our zero-shot approach with several methods that have used VPR datasets for training. Although EffoVPR-ZS was not trained on VPR task, it still achieves comparable results to the **trained** methods on three popular datasets. This success can be attributed to the robust features in DINOv2, specifically those selected from the $\mathcal{V}$ facet, combined with our mutual-NN matching and scoring. Figure 3b demonstrates this by showcasing a scenario where the original attention mistakenly focusing on an advertisement placed in front of a building. However, our method successfully identifies relevant key-points on the building itself, enabling correct image matching (even though a different advertisement is displayed on the gallery image).

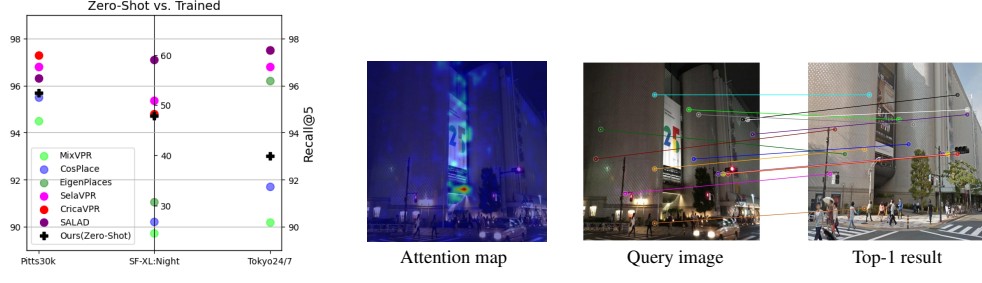

(a) Zero-shot vs. Trained          (b) EffoVPR zero-shot visualization

Figure 3: EffoVPR zero-shot. (a) Comparison of EffoVPR-ZS with other VPR trained methods. Our zero-shot approach shows comparable results. (b) Zero-shot success despite existing dynamic and irrelevant objects and strong visual change. Matching keypoints are indicated by colored lines. Although the pre-trained DINOv2 initially has its strongest attention on the distracting temporal advertisement, EffoVPR effectively identifies correct keypoints for successful matching.

## 4.2 Comparison with State-of-The-Art

In this section, we compare our single stage (EffoVPR-G) and two-stage methods (EffoVPR-R) with previous state-of-the-art including the recent works of (Lu et al., 2024a; Izquierdo & Civera, 2024; Lu et al., 2024b; Ali-bey et al., 2024). SelaVPR and R2Former were trained on a combination of Pitts30k and MSLS while CricaVPR, SALAD and BoQ were trained on GSV-Cities, and Cosplace and EigenPlaces on SF-XL (similar to ours). We show in Table 2 the global retrieval results (without

re-ranking) with Recall@1 on five different benchmarks. EffoVPR-G achieves SoTA performance on three out of five datasets, while being ranked second on the other two. This highlights the effectiveness of the single global representation learned by our method. Notably, it achieves $+2.9\%$ on Tokyo24/7, $+2.8\%$ on the challenging Nordland dataset that exhibits extreme seasonal changes, and $+3.2\%$ on the hold-out MSLS-challenge dataset. We further demonstrate the performance of our global feature learning with reduced dimensions of 256D and even 128D, significantly decreasing the memory footprint and enabling efficient searches within a considerably larger gallery. The findings indicate only a marginal degradation in performance with lower-dimensional features, while achieving parity with the SALAD on Tokyo24/7 using 128D, compared to 8,448D, feature size (a 66-fold reduction in dimensionality). Figure 1 illustrates this quality on Tokyo24/7 and the hold-out MSLS-challenge, showing our top performing results even with 128D feature size. Results on more datasets can be found in the Appendix. The strong impact of our finetuning process including the last five layers, is visualized in Figure 4.

Table 2: Comparison with our single stage method - Recall@1 performance. Two-stage methods are marked with †, and present 1st-stage performance (for fair comparison). The best results are highlighted in **bold** and the second is underlined. We present results from EffoVPR with three different feature dimensions.

| Method | Dim | Pitts30k | Tokyo24/7 | MSLS-val | MSLS-chall. | Nordland |
|---|---|---|---|---|---|---|
| CosPlace | 512 | 88.4 | 81.9 | 82.8 | 61.4 | 66.5 |
| MixVPR | 4096 | 91.5 | 86.7 | 88.2 | 64.0 | 58.4 |
| R2Former [†] | 256 | 76.3 | 45.7 | 79.3 | 56.2 | 50.9 |
| EigenPlaces | 2048 | 92.5 | 93.0 | 89.1 | 67.4 | 71.2 |
| SelaVPR[†] | 1024 | 90.2 | 81.9 | 87.7 | 69.6 | 72.3 |
| CricaVPR | 4096 | **94.9**[*] | 93.0 | 90.0 | 69.0 | 90.7 |
| SALAD | 8448 | 92.4 | 94.6 | **92.2** | 75.0 | 76.0 |
| BoQ | 16384 | 92.5 | 91.1 | 90.7 | 73.3 | 83.4 |
| EffoVPR-G[†] | 1024 | 94.8 | **97.5** | 90.9 | **78.2** | **93.5** |
| EffoVPR-G[†] | 256 | 93.8 | **95.9** | 90.4 | **75.6** | 79.7 |
| EffoVPR-G[†] | **128** | 92.6 | **94.6** | 88.2 | 73.8 | 70.4 |

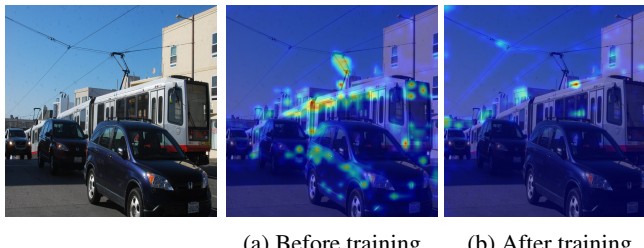

(a) Before training    (b) After training

Figure 4: Attention map visualization: pre-trained DINOv2 focuses on irrelevant foreground objects *e.g.* vehicles. Whereas attentions of EffoVPR after training are shifted to scene layout and building structures such as cables and windows.

Next, we showcase the comprehensive performance of our two-stage approach (EffoVPR-R) in Table 3. EffoVPR-R achieves top performance across all datasets, taking the second place only on Pitts30K-R@1, with very close result. Note that CricaVPR reports using Pitts30k as a validation set, which may have contributed to the improved results on this dataset. However, EffoVPR-R demonstrates notable improvements, particularly evident in Tokyo24/7, where it achieves a remarkable increase in R@1 from 94.6% to 98.7% and in MSLS-challenge (from to 75.0% to 79.0%). These results underscore the generalization capability of our approach, demonstrating its resilience in handling significant variations between query and gallery images, such as viewpoint discrepancies (as seen in Pitts30k) and changes in illumination (in Tokyo24/7), across a diverse range of locations. Following the common practice, we report EffoVPR-R re-ranking performance over top-100 candidates retrieved in the first-stage ($K = 100$), however we achieve SoTA R@1 results even with a low

Table 3: Comparison to state-of-the-art methods on four benchmarks. The bests results are highlighted in **bold** and the second is underlined. Two-stage methods are marked with †.

| Method | Pitts30k | | | Tokyo24/7 | | | MSLS-val | | | MSLS-challenge | | |
|---|---|---|---|---|---|---|---|---|---|---|---|---|
| | R@1 | R@5 | R@10 | R@1 | R@5 | R@10 | R@1 | R@5 | R@10 | R@1 | R@5 | R@10 |
| NetVLAD | 81.9 | 91.2 | 93.7 | 60.6 | 68.9 | 74.6 | 53.1 | 66.5 | 71.1 | 35.1 | 47.4 | 51.7 |
| SFRS | 89.4 | 94.7 | 95.9 | 81.0 | 88.3 | 92.4 | 69.2 | 80.3 | 83.1 | 41.6 | 52.0 | 56.3 |
| Patch-NetVLAD† | 88.7 | 94.5 | 95.9 | 86.0 | 88.6 | 90.5 | 79.5 | 86.2 | 87.7 | 48.1 | 57.6 | 60.5 |
| CosPlace | 88.4 | 94.5 | 95.7 | 81.9 | 90.2 | 92.7 | 82.8 | 89.7 | 92.0 | 61.4 | 72.0 | 76.6 |
| MixVPR | 91.5 | 95.5 | 96.4 | 86.7 | 92.1 | 94.0 | 88.2 | 93.1 | 94.3 | 64.0 | 75.9 | 80.6 |
| R2Former† | 91.1 | 95.2 | 96.3 | 88.6 | 91.4 | 91.7 | 89.7 | 95.0 | 96.2 | 73.0 | 85.9 | 88.8 |
| EigenPlaces | 92.5 | 96.8 | 97.6 | 93.0 | 96.2 | 97.5 | 89.1 | 93.8 | 95.0 | 67.4 | 77.1 | 81.7 |
| SelaVPR† | 92.8 | 96.8 | 97.7 | 94.0 | 96.8 | 97.5 | 90.8 | 96.4 | 97.2 | 73.5 | 87.5 | 90.6 |
| CricaVPR | **94.9** | 97.3 | 98.2 | 93.0 | 97.5 | 98.1 | 90.0 | 95.4 | 96.4 | 69.0 | 82.1 | 85.7 |
| SALAD | 92.4 | 96.3 | 97.4 | 94.6 | 97.5 | 97.8 | 92.2 | 96.2 | 97.0 | 75.0 | 88.8 | 91.3 |
| BoQ | 92.5 | 96.4 | 97.4 | 91.1 | 95.9 | 96.5 | 90.7 | 94.7 | 95.8 | 73.3 | 82.9 | 86.2 |
| EffoVPR-R† | 93.9 | **97.4** | **98.5** | **98.7** | **98.7** | **98.7** | **92.8** | **97.2** | **97.4** | **79.0** | **89.0** | **91.6** |

number of candidates (from $K = 5$ onwards, see Tab. S4 in Appx). Note that EffoVPR matching method is highly expedient, with an average processing time of just 1 millisecond per match.

In Figure 5, we present a failure case of our zero-shot approach, which is resolved after fine-tuning. In this instance, both the query and gallery contain a visually identical vehicle (an SF cable car), which leads to incorrect matching. Although such instances are rare in general case in context of pedestrians or vehicles in the images, where the objects are commonly not identical, this example highlights a limitation of our zero-shot approach.

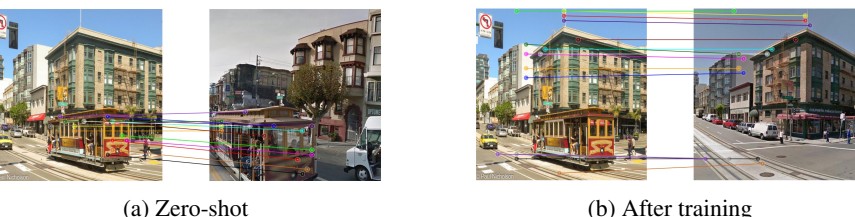

(a) Zero-shot                    (b) After training

Figure 5: Zero-shot vs trained: a failure case of our zero-shot approach, resolved after training.

Finally, we demonstrate the efficacy of our approach in the most challenging VPR benchmark scenarios by conducting experiments on six demanding datasets: Nordland (Sünderhauf et al., 2013), which includes extensive seasonal changes; AmsterTime (Yildiz et al., 2022), spanning over an extended time period; SF-Occlusion (Barbarani et al., 2023), that features queries with significant field-of-view obstructions; SF-Night (Barbarani et al., 2023), with severe illumination changes; and SVOX (Berton et al., 2021), with extreme weather and illumination variations. The results, detailed in Table 4, underscore the significant superiority of our method over previous approaches across these datasets. EffoVPR-R shows improvements of +4.3%, +0.8%, +7.9%, and +15%, +2% on Nordland, AmsterTime, SF-Occlusion, SF-Night, and SVOX-Night respectively and comparable results on SVOX-Rain. This demonstrates the high versatility of our model, which can handle extreme variations even when trained without seasonal or day-to-night changes. Figure 1 shows an examples of this case. We attribute this robustness primarily to the combination of our training method and specific re-ranking strategy over the DINOv2 model. We conduct an extensive ablation study on various hyperparameters and aspects of our approach in the Appendix.

Figure 6 qualitatively highlights the superior performance of our method. While other methods fail in challenging scenarios, such as viewpoint changes, seasonal variations, illumination differences, and severe occlusions, EffoVPR demonstrates high robustness against these challenges.

Table 4: Comparison (R@1) to SoTA methods on more challenging datasets.

| Method | Nordland | Amster Time | SF-XL Occlusion | SF-XL Night | SVOX Night | SVOX Rain |
|--------|----------|-------------|-----------------|-------------|------------|-----------|
| EigenPlaces | 71.2 | 48.9 | 32.9 | 23.6 | 58.9 | 90.0 |
| SelaVPR | 85.2 | 55.2 | 35.5 | 38.4 | 89.4 | 94.7 |
| CricaVPR | _90.7_ | _64.7_ | 42.1 | 35.4 | 85.1 | 95.0 |
| SALAD | 76.0 | 58.8 | _51.3_ | _46.6_ | _95.4_ | **98.5** |
| BoQ | 83.4 | 51.3 | 36.8 | 26.8 | 86.5 | 95.5 |
| EffoVPR-R | **95.0** | **65.5** | **59.2** | **61.6** | **97.4** | _98.3_ |

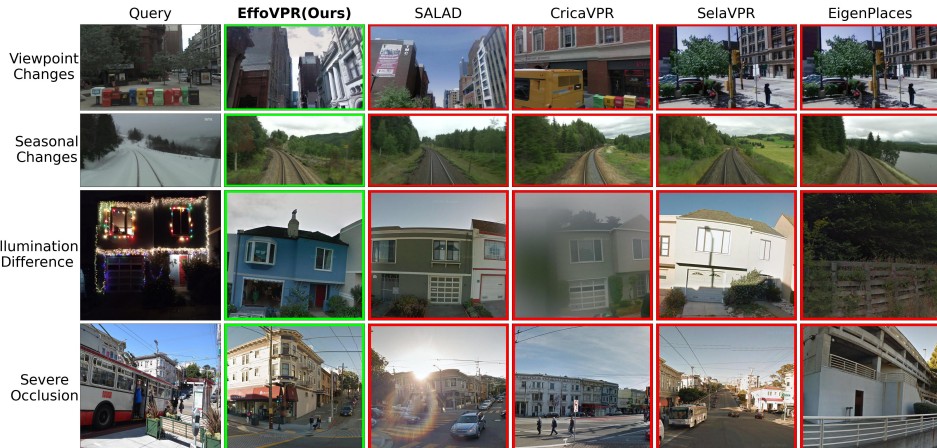

Figure 6: Qualitative comparison to SoTA Methods with challenging examples.

## 5 ABLATION STUDY

We conduct a comprehensive ablation study using two different datasets to analyze the key parameters of our EffoVPR method. Key findings are summarized below, with detailed results provided in the appendix. Analyzing the different ViT self-attention facets, we show that Value facet ($\mathcal{V}$) offers the most effective local features for VPR re-ranking (Table S3). Moreover, the selection of the layer for feature extraction is important, with the $n-1$ layer delivering the best performance, whereas using the final layer $n$ leads to a decline in results (Table S1). In Table S4 we show that our re-ranking stage remains effective even with as few as five candidates, achieving SoTA results, highlighting the method's efficiency and effectiveness. Fine-tuning only the last five layers of the backbone network proves optimal (Figure S1). Although EffoVPR achieves state-of-the-art results with lower resolutions, increasing image resolution up to $504$px slightly improves performance (Table S7).

## 6 SUMMARY

In this paper, we present both single-stage and two-stage approaches for VPR that leverage the internal layers of a foundation model for pooling and re-ranking. We challenge the necessity of recent complex architectural modifications, particularly the reliance on external pooling methods that lead to extended feature lengths in pursuit of high performance. By utilizing the model's existing pooling layers, we demonstrate state-of-the-art results with impressive feature compactness. We introduce a novel matching method for re-ranking based on self-attention facets of a foundation model, which proves effective even in a zero-shot setting.

Demonstrating strong robustness to various appearance changes while maintaining compact features, EffoVPR offers a promising solution for tackling the VPR task in real-world, large-scale applications.

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

# Appendix

## A    DATASETS

We evaluate EffoVPR performance across a large number of datasets, to underscore its top-performance in variable scenarios and cities. Following prior work, we have used (Berton et al., 2022c) open-source code for downloading and organizing datasets, to ensure maximum reproducibility. In the following we shortly describe each of the datasets.

### A.1    DATASETS SUMMARY

**AmsterTime** (Yildiz et al., 2022) consists of $1,231$ image pairs from Amsterdam, Holland, exhibiting long-term changes. The queries are historical grayscale images, where for each query there is a reference of a modern-day photo which represents the same place. The pairs curated by human experts, and provide multiple challenges over different viewpoints and cameras, color vs grayscale and long-term changes.

**Eynsham** (Cummins & Newman, 2009) is a collection of a car street-view camera, capturing photos around the same route of Oxford countryside twice. The grayscale images are divided to $23,935$ queries and $23,935$ gallery.

**Mapillary Street-Level Sequences (MSLS)** (Warburg et al., 2020) is image and sequence-based VPR dataset. The dataset consists of more than 1.6M geo-tagged images collected during over seven years from 30 cities, in urban, suburban, and natural environments. There are 3 non-overlap subsets - a training set, validation (MSLS-val), and withheld test (MSLS-challenge). MSLS-val and MSLS-challenge provide various challenges, including viewpoint variations, long-term changes, and illumination and seasonal changes.

**Nordland** (Sünderhauf et al., 2013) was collected by a mounted camera on the top of a riding train in the Norwegian countryside, presenting rural and natural scenes. The data collected over the same route across four seasons, providing seasonal and illumination variability. Following (Berton et al., 2022c; Sünderhauf et al., 2013) we use the post-processed versions of winter as queries and summer as database, determining correct localization by retrieval of an image that is in less than 10 frames away. This dataset consists of $27,592$ query images and $27,592$ gallery images

**Pittsburgh30k** (Arandjelovic et al., 2016) is collected from Google Street View 360° panoramas of downtown Pittsburgh, split into multiple images. Ensuring queries and gallery were taken in different years, it provides 3 splits - a training set, validation and test. Pitts30k-test consists of 10k gallery images and 6816 queries. Pitts250k consists of 8280 queries including these of Pitts30k, and its gallery size is $83,952$.

**San Francisco Landmark (SF-R)** (Chen et al., 2011) is a dataset from downtown San Francisco, which provides viewpoint variations. It presents a collection of $598$ of smartphone camera queries and gallery of $1,046,587$ images.

**San Francisco eXtra Large (SF-XL)** (Berton et al., 2022a; Barbarani et al., 2023) is an enormous dataset covering the whole city of San Francisco. it consists of a training set, which includes also raw 360° panoramas, a small validation set of $7,983$ queries and $8,015$ gallery images, and a test gallery of $2,805,840$ images.
There are four sets of queries:
*SF-XL-v1* (Berton et al., 2022a) consists of $1,000$ queries curated from Flickr, and provides viewpoint and camera variations, illumination changes and even some occlusions.
*SF-XL-v2* (Berton et al., 2022a) is the queries of San Francisco Landmark (SF-R).
*SF-XL-Night* (Barbarani et al., 2023) is a collection of 466 Flickr images of night scenes from San-Francisco. It provides viewpoint variations and very-challenging illumination changes.
*SF-XL-Occlusion* (Barbarani et al., 2023) is a collection of 76 Flick images from the city of San Francisco, which suffers from severe occlusions, mostly by vehicles and crowd.

**SPED** (Chen et al., 2018) is a collection of surveillance cameras images consists of 607 pairs of queries and gallery, captured accros time. It provides challenging viewpoint with seasonal and illumination changes.

**St Lucia** (Milford & Wyeth, 2008) is a collection of a nine videos of car-mounted camera from the St Lucia suburb of Brisbane. Following (Berton et al., 2022c) open-source code, we select the first and last videos as queries and database, and sample one frame every 5 meters of driving. The gallery consists of $1,549$ images and there are $1464$ query images.

**SVOX** (Berton et al., 2021) is a dataset which presents multiple weather conditions VPR challenge. It consists of $17,166$ gallery images, of the city of Oxford. The queries were collected from the Oxford RobotCar dataset (Maddern et al., 2017), providing multiple weather conditions queries sets, such as night (823 queries), overcast (872 queries), rainy (937 queries), snowy (870 queries) and sunny (854 queries).

**Tokyo 24/7** (Torii et al., 2015) is a dataset from downtown Tokyo, which provides viewpoint changes and challenging illumination variations. It consists of a gallery of $75,984$ images, and a collection of $315$ smartphone camera queries from $185$ places. Each place is portrayed by three photos - one taken during the day, one at sunset and one at night.

## A.2 TRAIN DATASET

Following VPR classification methods, Eigneplaces and CosPlace, we train on SF-XL (Berton et al., 2022a) while other studies (Ali-bey et al., 2022; Izquierdo & Civera, 2024; Lu et al., 2024a; Ali-bey et al., 2023) train on GSV-Cities (Ali-bey et al., 2022) or combinations of Pittsburgh30k (Arand-jelovic et al., 2016) and MSLS (Warburg et al., 2020; Arandjelovic et al., 2016; Lu et al., 2024b; Zhu et al., 2023), including a large mixture of different cities around the world (introducing higher variability). Note that similar to EigenPlaces, our approach is designed for training on panoramas with heading information, and requires slicing them for lateral and frontal views, which cannot be applied other training datasets.

## B ADDITIONAL ABLATION STUDIES

We conduct extensive experiments on two different datasets to ablate over key-components of our EffoVPR method.

**Re-ranking features**:

We explore various configurations for selecting features in the re-ranking stage. Our initial focus is on the choice of the layer from which features are extracted. Table S1 demonstrates that extracting features from the $n-1$ layer yields the most significant enhancement in overall performance. Generally, employing re-ranking with any layer, except of the last layer, improves results compared to omitting re-ranking entirely (*i.e.*, relying solely on the global feature from the first stage). Subsequently, upon extracting the Q, K, V components from the chosen layer, we find that the Value set ($\mathcal{V}$) represents the most effective local features for re-ranking, as detailed in Table S3. We ablate in Table S2 the impact of our two thresholds, the Attention Map threshold $T_1$ and the threshold on the countable local feature matching score threshold $T_2$.

Table S 1: **Ablation study on the choice of the layer for the re-ranking stage**. We find the $n-1$ layer to be the optimal for re-ranking feature extraction. Notably, the last layer $n$ is ineffective and downgrades global performance. Results (in %) are the R@1.

| Dataset | Global | n-5 | n-4 | n-3 | n-2 | n-1 | n |
|---|---|---|---|---|---|---|---|
| MSLS-val | 90.9 | 90.3 | 90.9 | 92.0 | 92.3 | **92.8** | 88.2 |
| Tokyo-24/7 | 97.5 | 98.1 | 98.1 | 98.1 | **98.7** | **98.7** | 97.1 |

Table S 2: **Ablation study on the impact of the thresholds**. Results (in %) are the R@1. $T_1$ is the Attention Map threshold and $T_2$ is the threshold on the countable local features matching score

|          | Tokyo24/7 | MSLS-val |
|----------|-----------|----------|
| no thr.  | 95.9      | 86.4     |
| $T_1$    | 97.1      | 91.5     |
| $+T_2$   | **98.7**  | **92.8** |

Table S 3: **Ablation on the choice of the local features**. Results (in %) are the R@1. *Query*, *Key* and *Value* are respectively $Q,K,V$ at Equation (1)

|           | Query | Key  | Value    |
|-----------|-------|------|----------|
| Tokyo24/7 | 96.5  | 96.8 | **98.7** |
| MSLS-val  | 89.7  | 90.1 | **92.8** |

**Number of Candidates to Re-rank**: The second re-ranking stage is applied to the top-K candidates retrieved during the global stage. Although common choice in literature is $K = 100$ (*e.g.* (Lu et al., 2024b; Zhu et al., 2023; Berton et al., 2022c)), we explore different choices of K, as detailed in Table S4. We achieve SoTA results even with $K = 5$. It is important to note that in some cases, an increase in K can introduce a greater number of "distractor" candidates, potentially leading to a decrease in performance. However, EffoVPR SoTA performance is consistent for all tested K's.

Table S 4: Re-ranking ablation. $K$ indicates re-ranking over top-$K$ results. We achieve SoTA results even with $K = 5$. Bold values indicate SoTA results.

| Top-K | Pitts30k | Tokyo24/7 | MSLS-val | Nordland | SF-XL-Occ. | SF-XL-Night | SPED |
|-------|----------|-----------|----------|----------|------------|-------------|------|
| K=5   | 94.2     | **97.8**  | **92.4** | **95.3** | **59.2**   | **61.6**    | 93.4 |
| K=10  | 94.2     | **98.1**  | **92.2** | **95.3** | **59.2**   | **61.2**    | 92.9 |
| K=15  | 94.1     | **98.4**  | **92.3** | **95.3** | **60.5**   | **60.3**    | 93.1 |
| K=20  | 94.0     | **98.4**  | **92.4** | **95.3** | **59.2**   | **60.3**    | 93.1 |
| K=50  | 93.9     | **98.7**  | **92.7** | **95.2** | **57.9**   | **60.9**    | 93.2 |
| K=100 | 93.9     | **98.7**  | **92.8** | **95.0** | **59.2**   | **61.2**    | 93.2 |

**Choice of Trainable Layers**: Figure S1 presents a few different sets of trainable layers in our backbone model. We find the vanilla fine-tuning of the entire model, end-to-end, that includes all layers, drastically harms the performance of EffoVPR-G in the global descriptors level. We attribute this decline to the fact that the DINOv2 backbone was trained on significantly larger datasets compared to those typically used in VPR. Subsequently, we establish that training only the last five layers represents a "sweet-spot", yielding peak performance. Both increasing or decreasing the number of trainable layers from this configuration leads to lower results.

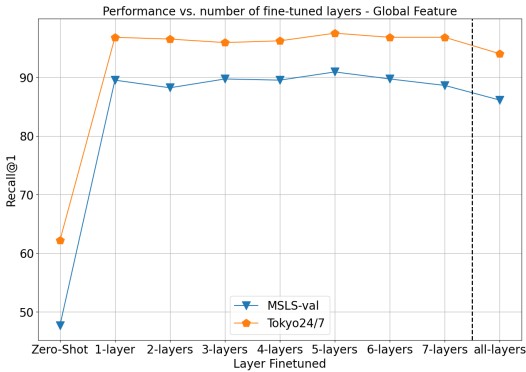

Fig. S 1: Ablation on the number of trainable layers
.

$T_1$ **and** $T_2$ **thresholds.** $T_1$ parameter shown in Fig.2 sets a threshold for selecting strong keypoints, while $T_2$ sets a threshold for identifying strong matching pairs. In practice, keypoints with scores ($s_i$

in Fig. 2) below $T_1$ are filtered out, and matching pairs with scores below $T_2$ are discarded. We set $T_1$ and $T_2$ values according to a validation set, and ablate the sensitivity for each value independently (while fixing the other) in Tables S5, S6.

Table S 5: **Ablation study on the value of $T_1$** ($T_2 = 0.65$). Results (in %) are the R@1.

| $T_1$ | 0.00 | 0.05 | 0.10 |
|---|---|---|---|
| Tokyo24/7 | 97.8 | **98.7** | 98.4 |
| MSLS-val | 89.1 | **92.8** | 92.4 |

Table S 6: **Ablation study on the value of $T_2$** ($T_1 = 0.05$). Results (in %) are the R@1.

| $T_2$ | 0.50 | 0.65 | 0.80 |
|---|---|---|---|
| Tokyo24/7 | 97.5 | **98.7** | 98.4 |
| MSLS-val | 91.8 | **92.8** | 91.5 |

**Image Resolution.** Different methods employed varying image resolutions. For instance, SALAD (Izquierdo & Civera, 2024) used a resolution of 322px. During inference, we used images with a resolution of 504px to enable more patches (of $14 \times 14$), particularly for re-ranking that uses local keypoint detection and matching. The increased number of patches primarily contributes to improved performance on more challenging datasets. However, image resolution impacts both inference time and memory footprint in the feed-forward path. Table S7 presents an ablation study on performance and feature extraction efficiency using intermediate resolutions of 224px and 322px across five datasets. EffoVPR-R consistently maintains state-of-the-art performance across various resolutions. Higher resolution enhances performance. Both GPU memory footprint and feature extraction time increase as the resolution rises.

Table S 7: **Ablation study on image resolution**. Results (in %) are the R@1 of EffoVPR-R

| Res. | GPU (GB) | Latency (s) | Tokyo 24/7 | MSLS val | Nord land | SF-XL Night | Amster Time |
|---|---|---|---|---|---|---|---|
| 224px | 2.33 | 0.018 | 98.4 | 91.1 | 92.7 | 57.5 | 65.1 |
| 322px | 2.38 | 0.028 | 98.4 | 92.8 | 94.3 | 61.2 | 65.3 |
| 504px | 2.57 | 0.067 | 98.7 | 92.8 | 95.0 | 61.6 | 65.5 |

# C   ADDITIONAL RESULTS

## C.1   PERFORMANCE VS. FEATURE COMPACTNESS

In Figure S2 we present performance comparison of our *global* feature (EffoVPR-G) versus feature dimensionality for more datasets. While the current leading methods achieve their performance using large features, EffoVPR demonstrates high performance even with an extremely compact feature size.

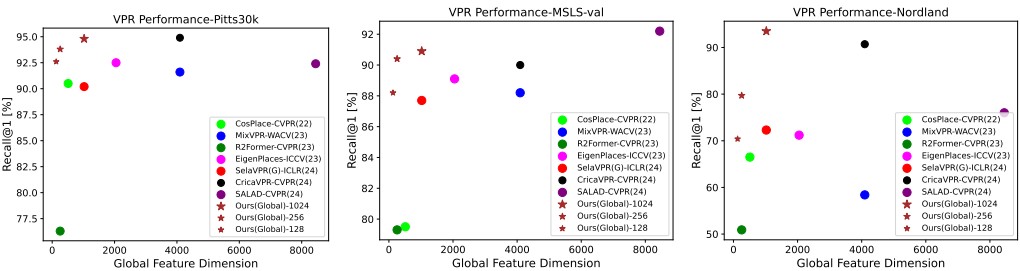

Fig.  S 2: Recall@1 performance of EffoVPR-G *global* feature versus feature dimensionality for more datasets.

## C.2 COMPARISON OF EFFOVPR MODEL VARIANTS

We further summarize the results of different variations of EffoVPR over five datasets in Table S8.

| Method | Pitts30k | Tokyo24/7 | MSLS-val | MSLS-chall. | Nordland |
|--------|----------|-----------|----------|-------------|----------|
| EffoVPR-ZS | 89.4 | 90.8 | 70.3 | 41.3 | 57.9 |
| EffoVPR-G | 94.8 | 97.5 | 90.9 | 78.2 | 93.5 |
| EffoVPR-R | 93.9 | 98.7 | 92.8 | 79.0 | 95.0 |

Table S 8: Recall@1 performance for different variations of EffoVPR.

## D VISUALIZATIONS

### D.1 ADDITIONAL ZERO-SHOT VISUALIZATIONS

In Figure S3 we show more visualizations of EffoVPR-ZS method. While the attention map of pre-trained DINOv2 doesn't focus on discriminative VPR elements, EffoVPR is able to fill the gap in zero-shot with local features matching. In the first row The pre-trained attention-map is mainly focused on temporal traffic signs and a far ad and almost not attend the building, and in the second row it is mainly focused on an insignificant back of a traffic sign. However EffoVPR method finds multiple local matches to the right geo-tagged image in the gallery.

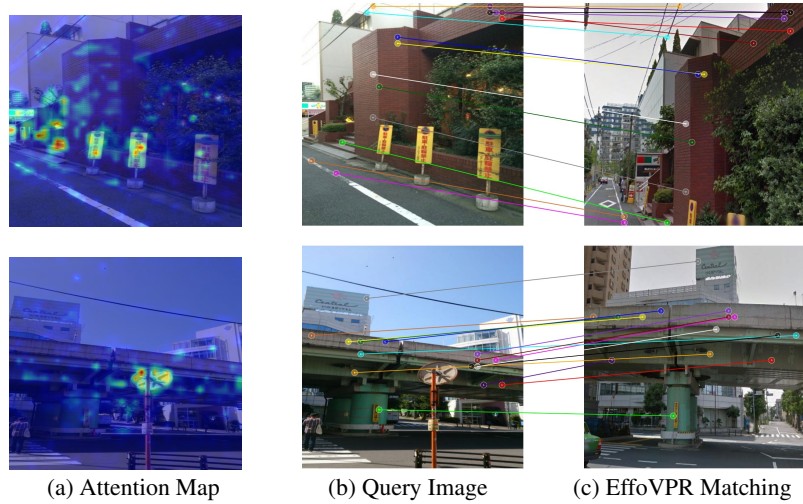

|            (a) Attention Map            |            (b) Query Image            |            (c) EffoVPR Matching            |

Fig. S 3: Additional EffoVPR-ZS zero-shot visualizations.

### D.2 ADDITIONAL RE-RANKING VISUALIZATIONS

Figure S4 exhibits EffoVPR-R local features matching invariability to highly challenging scenes with top-1 results. From the top left to the bottom right - to camera rotation, a nature scene, color variance across time (building renovation), tree matching, challenging day-time change with hardly noticed electric cables matching, night to day significant change.

### D.3 FAILURE CASES

Figure S5 presents several failure cases where EffoVPR-R produces incorrect predictions. In the first row, on the left, a cable car obscures most of the scene, and the model erroneously matches its eastern-style roof with a gallery image featuring the same cable car. On the right, a car and a tree obscure most of the scene, leaving insufficient features for accurate matching. As a result, the

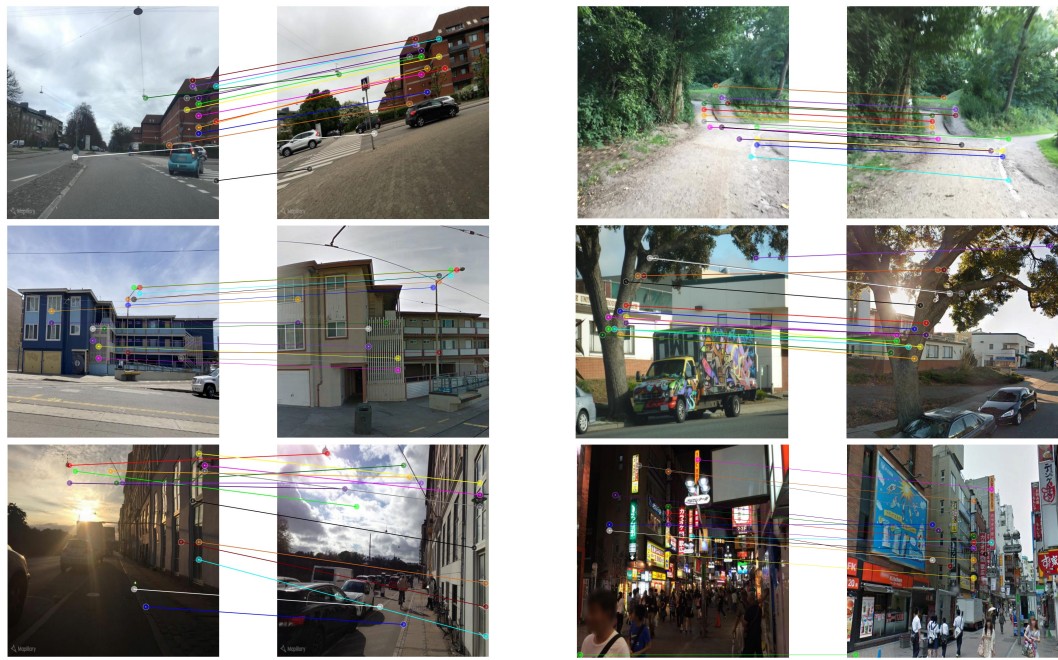

Fig. S 4: EffoVPR-R top-1 matching visualizations. For each pair matching, the left image is the query and the right is the top-1 result.

model incorrectly associates the tree and the white stripes of a pedestrian crossing with the wrong location. In the bottom row, a significant time-gap change is evident between the query and gallery images, making it challenging to identify the house of interest in the gallery, as it is depicted during a period of construction. Figure S6 illustrates a more common scenario observed in our results, where the model produces a visually accurate prediction that is nevertheless classified as incorrect. This occurs because the distance between the gallery image and the query image exceeds 25 meters, the threshold defined for the VPR task by previous studies.

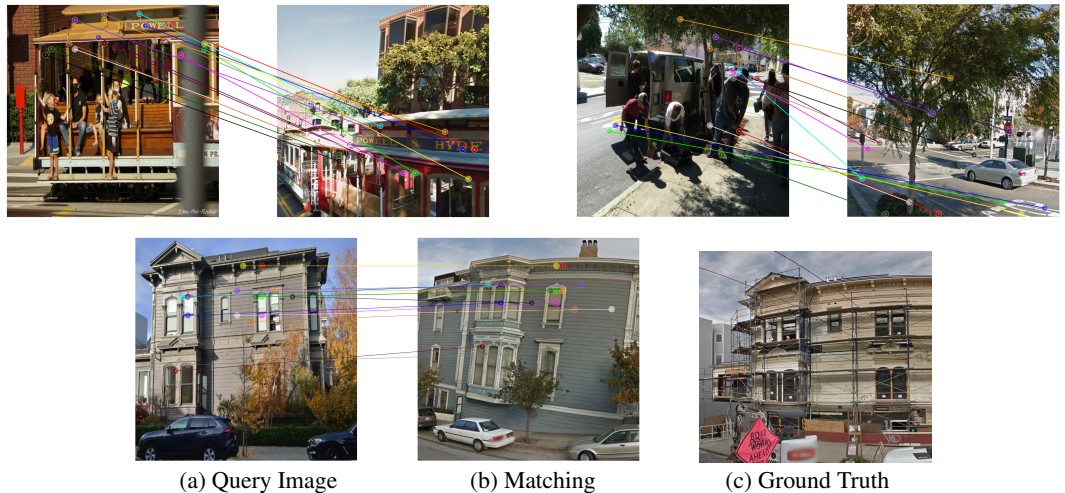

(a) Query Image        (b) Matching        (c) Ground Truth

Fig. S 5: EffoVPR-R failure cases stem from obscured views, lack of significant features and time gaps.. In the first row - for each pair matching, the left image is the query and the right is the top-1 result.

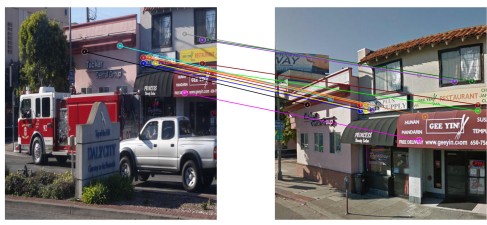 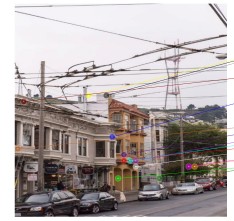 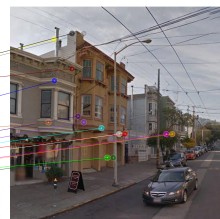

Fig. S 6: EffoVPR-R failure cases due to distance from the camera. For each pair matching, the left image is the query and the right is the top-1 result.

## E    IMPLEMENTATION DETAILS

We use ViT-L/14 as the backbone, initialized with pre-trained weights of DINOv2 with registers (Oquab et al., 2023; Darcet et al., 2023). We only train the last five layers of the backbone, which appeared to be most beneficial. We employ EigenPlaces's (Berton et al., 2023) group and class partitioning with its default hyper-parameters, and both lateral and frontal views, on the publicly available *SF-XL* street-view panoramas dataset (Berton et al., 2022a). We set an *AdamW* optimizer to the backbone, and an *Adam* optimizer to the classification heads, both with a constant learning rate of $1 \times 10^{-5}$. We train EffoVPR with a batch size of 16, for 25 epochs, on a single *NVIDIA-A100* node. We otherwise follow EigenPlaces training recipe. We choose the best epoch by *SF-XL* validation set, measuring Recall@1 global ranking performance. Given that ViT is independent of the image input size (provided it can be segmented into $14 \times 14$ patches), we evaluated using images sized $504 \times 504$, but trained on $224 \times 224$ images to expedite training. For benchmarking EffoVPR-G of the global feature, we report nearest-neighbors performance on normalized output class token. In the re-ranking stage we extract the $V$ self-attention facet from layer $n-1$ (with $n$ being the output layer), measure cosine similarity, we filter the features by class attention map with a threshold $\mathcal{T}_1 = 0.05$, and count only mutual nearest-neighbors with a score above the threshold $\mathcal{T}_2 = 0.65$.

### E.1    ADDITIONAL INFORMATION

#### E.1.1    ZERO-SHOT

In evaluating AnyLoc (Keetha et al., 2023), we tackle the significant memory requirements of its VLAD pooling algorithm by implementing an online clustering scheme. We observed that their recommendation for layer 31 outperformed layer 23. In our zero-shot evaluation, we assess EffoVPR-ZS method by extracting the $V$ features from layer $n-2$ to re-rank the top-100 candidates retrieved from the first-stage global [CLS] feature. In this framework, our performance is constrained by the first-stage Recall@100, achieving rates of 99.2%, 96.8%, 81.5%, and 78.1% on Pitts30k, Tokyo24/7, MSLS-Val, and Nordland, respectively.

#### E.1.2    BENCHMARKING

Generally, for consistent benchmarking, we adhere to (Berton et al., 2022c). In addition, we report the results of other methods in accordance with the evaluation choices of SelaVPR (Lu et al., 2024b) and CricaVPR (Lu et al., 2024a), including the specific versions of trained models utilized. For the recent state-of-the-art methods SelaVPR, CricaVPR and SALAD, we provide results from the original publications whenever available. When such results are not directly available, we utilize their code and published weights. Specifically for SelaVPR, which has two sets of weights (trained on Pitts30k and MSLS), we report the best-performing for each dataset. For BoQ, we evaluated by the model provided weights across all datasets, due to inconsistency in reported results in the paper, and project page and code.

### E.2    OTHER

We evaluate EffoVPR matching runtime and local features dimension by averaging matching function runtime and local features number on the Tokyo 24/7 dataset.

## F  MEMORY FOOTPRINT AND RUNTIME

One important aspect of the memory footprint in VPR stems from the size of the model's features. This directly affects the gallery size in RAM, influencing the efficiency of fast search and retrieval. In Table S9, we present a detailed breakdown of the memory footprint for our global ranking stage, comparing it to other methods. This includes an analysis based on a collection of 1 million images, demonstrated using the real-world SF-XL gallery, which covers the city of San Francisco with 2.8 million images. EffoVPR-G achieves high performance even with 128-dimensional features, requiring as little as 500MB per million images, and 1.34GB for the city of San-Francisco.

| Method | Features Dim. | 1M images (GB) | SF-XL Gallery (GB) |
|---|---|---|---|
| R2Former[†] | 256 | 0.98 | 2.68 |
| SelaVPR[†] | 1024 | 3.81 | 10.7 |
| CricaVPR | 4096 | 15.26 | 42.81 |
| SALAD | 8448 | 31.47 | 88.3 |
| BoQ | 16384 | 61.04 | 171.25 |
| EffoVPR-G[†] | 1024 | 3.81 | 10.7 |
| EffoVPR-G[†] | 256 | 0.98 | 2.68 |
| EffoVPR-G[†] | 128 | 0.49 | 1.34 |

Table S 9: Feature dimension and memory usage for different methods.

In Table S10, we evaluate memory footprint and runtime of our re-ranking stage comparing to R2Former and SelaVPR as reported in their corresponding paper.

| Method | Local Features Dim. | Mem. Footprint (GB) | Latency (s) | Tokyo24/7 R@1 | MSLS-Val R@1 |
|---|---|---|---|---|---|
| R2Former | $500 \times (128 + 3)$ | 0.025 | 0.202 | 88.6 | 89.7 |
| SelaVPR | $61 \times 61 \times 128$ | 0.182 | 0.085 | 94.0 | 90.8 |
| EffoVPR-R | $649 \times 1024$ | 0.254 | 0.035 | 98.7 | 92.8 |

Table S 10: Local features dimension, memory footprint, latency and performance for different methods. Different methods utilized different GPUs for runtime evaluation: R2Former runtime was measured using RTX A5000, SelaVPR with RTX 3090, and EffoVPR with A100.

## G  ADDITIONAL QUANTITATIVE RESULTS

To ensure comprehensiveness, the following Table S11 presents the complete results for datasets that were only partially presented in the main paper, as well as for some datasets that were previously omitted. Our method, EffoVPR, demonstrates SoTA performance on the majority of these datasets, and remains competitive with the SoTA on others.

Table S 11: Comparison to SoTA on more datasets

| Method | SPED | | | SF-R | | | SF-XL-v1 | | | SF-XL-v2 | | | SF-XL-Occ. | | |
| | R@1 | R@5 | R@10 | R@1 | R@5 | R@10 | R@1 | R@5 | R@10 | R@1 | R@5 | R@10 | R@1 | R@5 | R@10 |
|---|---|---|---|---|---|---|---|---|---|---|---|---|---|---|---|
| EigenPlaces | 70.2 | 83.5 | 87.5 | 89.6 | 94.3 | 95.3 | 84.1 | 89.1 | 90.7 | 90.8 | 95.7 | 96.7 | 32.9 | 48.7 | 52.6 |
| SelaVPR | 88.6 | 95.1 | 97.2 | 88.5 | 92.0 | 93.0 | 74.9 | 80.7 | 82.1 | 89.3 | 95.7 | 96.3 | 35.5 | 47.4 | 55.3 |
| CricaVPR | 91.3 | 95.2 | 96.2 | 88.6 | 94.0 | 95.7 | 80.6 | 87.6 | 89.8 | 90.6 | 96.3 | 97.7 | 42.1 | 52.6 | 57.9 |
| SALAD | 92.1 | 96.2 | 96.5 | 92.3 | 95.7 | **96.8** | 88.6 | 93.5 | 94.4 | **94.8** | 97.3 | **98.3** | 51.3 | 65.8 | 68.4 |
| BoQ | 86.2 | 94.4 | 96.1 | 91.1 | 94.1 | 95.3 | 83.7 | 87.8 | 89.2 | 92.8 | 95.0 | 96.3 | 36.8 | 47.4 | 52.6 |
| EffoVPR | **93.1** | **97.9** | **98.4** | **93.0** | **96.0** | 96.3 | **95.5** | **98.1** | **98.3** | 94.5 | **97.8** | 98.2 | **59.2** | **68.4** | **73.7** |

| Method | SF-XL-Night | | | Amster Time | | | SVOX | | | SVOX Night | | | SVOX Overcast | | |
| | R@1 | R@5 | R@10 | R@1 | R@5 | R@10 | R@1 | R@5 | R@10 | R@1 | R@5 | R@10 | R@1 | R@5 | R@10 |
|---|---|---|---|---|---|---|---|---|---|---|---|---|---|---|---|
| EigenPlaces | 23.6 | 30.7 | 34.5 | 48.8 | 69.5 | 76.0 | 98.0 | 99.0 | 99.2 | 58.9 | 76.9 | 82.6 | 93.1 | 97.8 | 98.3 |
| SelaVPR | 38.4 | 50.9 | 55.4 | 55.2 | 72.6 | 78.0 | 97.2 | 98.7 | 99.0 | 89.4 | 95.5 | 96.6 | 97.0 | 99.1 | 99.3 |
| CricaVPR | 35.4 | 48.3 | 53.4 | 64.7 | 82.5 | 87.9 | 97.8 | 99.2 | 99.3 | 86.3 | 95.3 | 96.6 | 96.7 | 99.0 | 99.0 |
| SALAD | 46.6 | 59.0 | 62.2 | 58.8 | 78.9 | 84.2 | 98.2 | 99.3 | 99.4 | 95.4 | 99.3 | 99.4 | 98.3 | **99.3** | 99.3 |
| BoQ | 26.8 | 36.5 | 40.6 | 51.3 | 71.7 | 78.1 | **98.8** | 99.4 | 99.5 | 86.5 | 94.4 | 96.4 | 97.9 | 99.2 | 99.3 |
| EffoVPR | **61.6** | **73.4** | **77.0** | **65.5** | **87.2** | **90.7** | 98.7 | **99.5** | **99.6** | **97.4** | **99.5** | **99.5** | **98.4** | **99.3** | **99.7** |

| Method | SVOX Rain | | | SVOX Snow | | | SVOX Sun | | | Sr. Lucia | | | Eynsham | | |
| | R@1 | R@5 | R@10 | R@1 | R@5 | R@10 | R@1 | R@5 | R@10 | R@1 | R@5 | R@10 | R@1 | R@5 | R@10 |
|---|---|---|---|---|---|---|---|---|---|---|---|---|---|---|---|
| EigenPlaces | 90.0 | 96.4 | 98.0 | 93.1 | 97.6 | 98.2 | 86.4 | 95.0 | 96.4 | 99.6 | 99.9 | **100.0** | 90.7 | 94.4 | 95.4 |
| SelaVPR | 94.7 | 98.5 | 99.1 | 97.0 | 99.5 | 99.5 | 90.2 | 96.6 | 97.4 | 99.8 | 100.0 | 100.0 | 90.6 | **95.3** | 96.2 |
| CricaVPR | 94.8 | 98.5 | 98.7 | 96.0 | 99.2 | 99.2 | 93.8 | 98.1 | 98.8 | 99.9 | 99.9 | 99.9 | 91.6 | 95.0 | 95.8 |
| SALAD | **98.5** | **99.7** | **99.9** | **98.9** | **99.7** | **99.8** | 97.2 | **99.4** | **99.7** | **100.0** | 100.0 | 100.0 | **91.6** | 95.1 | 95.9 |
| BoQ | 95.5 | 99.3 | 99.7 | 98.5 | 99.5 | 99.7 | 96.1 | 98.8 | **98.9** | **100.0** | 100.0 | 100.0 | 91.2 | 94.9 | 95.9 |
| EffoVPR | 98.3 | 99.6 | 99.6 | 98.7 | **99.7** | 99.7 | **97.7** | 99.3 | 99.4 | **100.0** | 100.0 | 100.0 | 91.0 | 95.2 | **96.3** |

