# OpenReview forum: "EffoVPR: Effective Foundation Model Utilization for Visual Place Recognition"
_ICLR.cc/2025/Conference — ICLR 2025 Poster_

### Official Review · Reviewer_KXCa · 2024-10-29

**Soundness:** 3
**Presentation:** 3
**Contribution:** 2
**Rating:** 8
**Confidence:** 5

**Summary:**

The paper presents a model for visual place recognition, specifically aimed at achieving robust zero-shot retrieval performance. The model uses a Vision Transformer (ViT) with DINOV2 weights as its backbone to identify nearest neighbors based on global feature similarity, followed by a mutual nearest neighbor (NN) re-ranking strategy to pinpoint the closest reference image using local features. For training, the Eigenplace paradigm is applied to promote view invariance in the ViT. Only the final layers of the backbone are fine-tuned, with the rest of the model kept frozen. Fine-tuning was performed on the SF-XL dataset, and testing was conducted on several benchmarks: Pitts30k, Tokyo24/7, MSLS-val, and MSLS-challenge, as well as more challenging datasets, including Nordland, Amsterdam Time, SF-XL Occlusion, SF-XL Night, SVOX Night, and SVOX Rain.  For contribution, EffoVPR proposed a novel reranking architecture that does not require training and works well even with a small number of candidates.  The paper also claims to demonstrate the lack of need for aggregate methods to capture local features

**Strengths:**

Performance:
EffoVPR achieves SOTA performance by performing better than previous recent SOTA models such as SALAD and EigenPlaces.  In particular, EffoVPR performs significantly better on the more challenging datasets
EffoVPR is able to fine tune on cities in SF XL but then perform very well on Nordland which is mainly rural and natural scenes taken from a train.
EffoVPR has very impressive zero shot performance compared to other zero shot methods and some trained models, showing effectiveness of reranking model
EffoVPR with only the first stage also performs very well
Architecture:
The architecture for EffoVPR is intuitive and simple to understand
Efficiency:
EffoVPR utilizing only reranking during evaluation makes training efficient and fast, as the reranking does not need to be trained
Experimentation:
Paper presents very detailed ablation studies on the choice of layer for reranking, the different thresholds, which layer to choose reranking features for, etc.
Papers shows the model being performed on a wide range of datasets.

**Weaknesses:**

Performance:
Model does not perform better than other models by a significant margin on the less challenging datasets.  However, the less challenging datasets are already inferred very well by the other datasets so there’s not as much room for improvement.
Paper is not entirely clear how much the reranking contributes to performance compared to DINOV2
Novelty:
In contribution, the paper claimed “We introduce a novel image matching method for VPR that relies on keypoint detection and descriptor extraction from the intermediate layers of a foundation model”.  However, R2Former also uses filtered local features from intermediate layers from the VIT backbone. Perhaps need to change wording to reflect true level of novelty.

The paper mentions reduced memory footprint, but only shows dimension of global feature and not other aspects such as parameter size (ex. R2Former uses VIT-S whereas EffoVPR uses VIT-L but is not reflected anywhere on paper)

Training:
T1, T2, along with which layer to get feature map all being fixed hyperparameters means there’s more manual tuning involved to achieve optimal results

**Questions:**

The paper mentions time evaluation, but only shows time evaluation on the first step and not the reranking.  From the look of the model, the time of reranking doesn’t seem to be particularly time efficient.  How fast is the reranking?

The paper mentioned memory efficiency and reduced memory footprint.  Quantitatively, how much memory consumption does the modal overall take compared to the other models?

On the ablation on T1 and T2, table only shows the result with different T1 and different T2, but didn’t specify what the other filter value is (ex. On table S5, When showing performance when T1 equals 0.05 and 0.10, it didn’t specify what T2 is when evaluating).  What is the value of T2 when evaluating for table S5 and what is the value of T1 when evaluating for table S6?

How well does the model do on the Pittsburgh 250k dataset?

Are there experimental results of EffoVPR trained on other datasets other than SFXL?

In the Eigenplaces paper which the training process of EffoVPR seems to be heavily based on, the backbone was completely trained.  How well would the model perform if every layer of the backbone was trained?

Paper mentions that only the “final layers” are trained.  What are the final layers that are trained?

Paper calls itself a “foundational model”, which should be able to be used for multiple tasks, yet only same view matching capability is shown in the paper.  Can the model be used for other tasks like cross view matching or detection?

Could the global descriptor be used also during the reranking phase to further filter out the nearest neighbors, perhaps to only neighbors that belong in the same predicted cell?

How well does the model perform with anyloc as backbone rather than DINOV2?

---

> ### Author Response · Authors · 2024-11-22
> **Response to Review Part 1/2**
>
> We thank the reviewer for the positive feedback, acknowledging our SoTA performance, robustness (performing well on datasets from a distinct domain), impressive zero-shot performance, utilizing only reranking during evaluation makes training efficient and fast and having detailed ablation studies
>
>  **W1: datasets so there’s not as much room for improvement...**
>
>  **WA1:** We thank the reviewer for the comment. We compare our method on the common benchmarks of VPR. Additionally, we demonstrate our model’s superior performance on challenging datasets such as SF-XL Night, SF-XL Occlusion, Nordland and others. EffoVPR outperforms existing methods by a substantial margin, achieving improvements of +15.0, +7.9, and +4.3 in Recall@1, respectively.
>
>  **W2: How much the reranking contributes to performance compared to DINOv2**
>
>  **WA2:** Please note that the first row in Table 1 indicates the zero-shot DINOv2 results without re-ranking, while EffoVPR-ZS (zero-shot) indicates DINOv2 with our re-ranking strategy. We show these results below.
>
>  |Method|Pitts30k|Tokyo24/7|MSLS-val|MSLS-chall.|Nordland|
> |------|--------|---------|--------|-----------|--------|
> |DINOv2|78.1|62.2|47.7|23.9|33.0|
> |+Reranking|89.4|90.8|70.3|41.3|57.9|
>
>  **W3: R2Former also uses filtered local features from intermediate layers from the VIT backbone...**
>
>  **WA3:**  To the best of our knowledge, R2Former does not utilize local features from the intermediate layers of the ViT backbone but instead relies on the **output layer**. Specifically, R2Former leverages the attention map from the final layer to incorporate the most activated output tokens for its classification model. Our approach leverages different feature, K,Q and particularly V-facets from the model’s intermediate layers, that does not require any additional trained modules (in contrast to R2Former).
>
>  **W4: Memory footprint…**
>
>  **WA4:** Thank you for your comment. We provide a breakdown of the memory footprint in the table below, which has been added to the revised version of the paper. For demonstration purposes, we present the memory consumption for 1 million images and the entire SF-XL gallery.
>
>  |Method|Features Dim.|1M images (GB)|SF-XL Gallery (GB)|
> |------|-------------|--------------|------------------|
> |R2Former|256|0.98|2.68|
> |SelaVPR*|1024|3.81|10.7|
> |CricaVPR|4096|15.26|42.81|
> |SALAD|8448|31.47|88.3|
> |BoQ|16384|61.04|171.25|
> |EffoVPR-G|1024|3.81|10.7|
> |EffoVPR-G|256|0.98|2.68|
> |EffoVPR-G|128|0.49|1.34|
>
> Please note that we specify our ViT-L backbone in the implementation details section (see line 1012, in the original submission).
>
> **W5: More manual tuning involved…**
>
>  **WA5:** Thank you! While our search process was somewhat manual, we believe there is potential for a better optimization to further enhance performance. We would like to emphasize that throughout all tests and benchmarks our hyperparameters remain fixed and we provide an analysis of the results’ sensitivity to hyperparameters in the ablation study.
>
>  **Q1: Time evaluation and memory footprint**
>
>  **A1:** Thank you for your comment. For the memory footprint in the first stage please see our answer above. To address other concerns, we summarize the runtime comparison with several other methods as reported in their corresponding papers.
>
>  |Method|Local Features Dim.|Mem. Footprint (GB)|Latency (s)|Device|Tokyo 24/7 R@1|MSLS-Val R@1|
> |------|-------------------|-------------------|-----------|------|-------------|------------|
> |R2Former|500x(128+3)|0.25|0.202|RTX A5000|88.6|89.7|
> |SelaVPR|61x61x128|0.182|0.085|RTX 3090|94.0|90.8|
> |EffoVPR-R|649x1024|0.254|0.035|A100|98.7|92.8|
>
> **Q2: More details for Table S5**
>
>  **A2:** Thank you. The table shows the values where the counterpart parameter is set to its selected value. We have added this information to the paper.
>
>  **Q3: Results on Pittsburg 250K**
>
> **A3:** Thank you. We have conducted this test. Please see below the requested results:
>
> |Pitts-250K|SALAD|BoQ|EffoVPR|
> |----------|:---:|:-:|:-----:|
> |R@1|95.1|95.0|**97.0**|

---

> ### Author Response · Authors · 2024-11-22
> **Response to Review Part 2/2**
>
> **Q4: Train other datasets except SFXL**
>
> **A4:** Thank you for your comment. We discuss this in the appendix, section A.2. We follow the common practice, in recent VPR methods e.g. CosPlace, EigenPlaces, SelaVPR, CricaVPR, SALAD, which utilize a single dataset for training. Our training approach is based on EigenPlaces, which necessitates a dense coverage of 360 panoramas with orientation (heading) metadata. Other available training datasets lack the necessary metadata, preventing us from using them for training.
>
> **Q5: How well would the model perform if every layer of the backbone was trained?**
>
> **A5:** Please note that we specify and discuss this point in Table S1 of Appendix B, showing that training all layers is less effective.
>
> **Q6: What are the trained final layers**
>
> **A6:** We fine-tuned the last five layers as specified in line 522 (original submission) in the Ablation Study, and further discussed in Appendix B.
>
> **Q7: Paper calls itself a “foundational model”, Can the model be used for other tasks like cross view matching or detection?**
>
> **A7:** Please note that we follow the common terminology in the recent foundation based VPR methods, and also refer to our model as a “foundation based model”. This implies that the model backbone is a pretrained foundation model. Similar to other VPR methods such as SelaVPR, CricaVPR, SALAD, our task is for the same-view rather than cross-view.
>
> **Q8: Using global descriptor be used also during the reranking... only neighbors that belong in the same predicted cell.**
>
> **A8:** Thank you! This is an interesting idea. However, this approach leverages additional information from the geo-tags in the gallery, which is not typically used/allowed in standard comparisons. Furthermore, in certain competitions, such as MSLS-challenge, the geo-tags are sealed. Nonetheless, since this information can be available in practical scenarios, it presents a promising direction for future research.
>
> **Q9: How well does the model perform with anyloc as backbone rather than DINOV2?**
>
> **A9:** Please note that AnyLoc is not a backbone, but a method based on DINOv2 for a zero-shot VPR. We compare and show the superiority of EffoVPR in zero-shot setting over AnyLoc in Table 1.

---

> > ### Comment · Reviewer_KXCa · 2024-11-25
> > **Response to rebuttal**
> >
> > I have reviewed the rebuttal and satisfy with the response and increasing score.

---

> > > ### Author Response · Authors · 2024-11-28
> > >
> > > We want to thank the reviewer for his time and effort in reviewing our paper and rebuttal. We are happy that you found our response satisfactory, and increased your score. Thank you!

---

### Official Review · Reviewer_P6DW · 2024-11-01

**Soundness:** 3
**Presentation:** 3
**Contribution:** 3
**Rating:** 8
**Confidence:** 3

**Summary:**

The paper introduces a method that uses DINOv2 foundation model to address the Visual Place Recognition task. This is achieved by retrieving the top-K most similar candidates in the first stage by measuring the similarity of (finetuned) DINOv2’s [CLS] tokens between the query and the candidate images. In the second stage, the proposed approach re-ranks the retrieved candidates using local features computed using the Queries, Keys and Values from the self-attention module of an intermediate DINOv2 layer. The method outperforms other state-of-the-art zero-shot approaches, as well as supervised methods when the last layers of the DINOv2 network are finetuned. Moreover, the proposed method achieves results competitive with the supervised methods even when using only the first stage of the method.

**Strengths:**

- S1: The paper is clear and easy to read.

- S2: Simple, yet effective and robust method that improves the performance over the DINOv2 baseline and other state-of-the-art methods, while reducing the required dimensionality of the used features.

- S3: The method finetunes DINOv2 to achieve better performance but is still effective even in a zero-shot setting.

**Weaknesses:**

- W1 Even though the results of the three proposed variants of the method (EffoVPR-ZS, EffoVPR-G and EffoVPR-R) are provided, the paper is missing their comparison in a single table, to directly see how large is the effect of finetuning and re-ranking. It would be good to see this comparison in a single table to understand (and compare) the direct benefits of the different components of the proposed approach.

- W2 The notation for Queries, Keys and Values and corresponding local features would be more readable if the layer index would be consistently used always as superscript or always as subscript ($V_l$ versus $v^l_i$). Alternatively, since the Queries, Keys and Values are always taken from the same layer, maybe the layer index could be completely omitted for better readability.

- W3 What are the failure cases and limitations of the main presented method, EffoVPR-R? The paper contains only failure cases for the zero-shot setting.

**Questions:**

- Q1: The paper shows that EffoVPR-R achieves SOTA results even with K=5. I am curious, how much does this change for the zero-shot setting. I.e., when the last layers of DINOv2 are not finetuned, is it also sufficient to use K=5, or would K need to be increased?

- Q2: It should be clearly stated that the ablations in the appendix we done using the EffoVPR-R. Would it be possible to confirm this?

- Q3: The re-ranking stage requires storing local features for the re-ranked images. It would be good to add discussion and analysis of memory requirements for the proposed approach and in particular the memory requirements to store the local features for spatial re-reranking.

---

> ### Author Response · Authors · 2024-11-22
> **Response to Review**
>
> Thank you for acknowledging our method as simple yet effective, and for finding our paper clear and easy to read.
>
>  **W1: comparison in a single table**
>
>  **WA1:** Thank you for this comment. Below we present this table that we also added to a revised version of the paper (Appendix C).
>
>  |Method|Pitts30k|Tokyo24/7|MSLS-val|MSLS-chall.|Nordland|
> |------|--------|---------|--------|-----------|--------|
> |EffoVPR-ZS|89.4|90.8|70.3|41.3|57.9|
> |EffoVPR-G|94.8|97.5|90.9|78.2|93.5|
> |EffoVPR-R|93.9|98.7|92.8|79.0|95.0|
>
> **W2: The notation for Keys, Queries and Values**
>
>  **WA2:** Thank you for this constructive remark. We have now omitted the index $l$ in the revised formulation for clarity and explained the dependency to layer in the text.
>
>  **W3: Failure cases**
>
>  **WA3:** Thank you! We have added several failure cases, categorizing and discussing these cases in the Appendix (section D.3) of the revised version.
>
>  **Q1: Performance for different K values in zero-shot setting...**
>
>  **A1**: Thank you for your comment. We have conducted an ablation study on EffoVPR-ZS Recall@1 performance across different K values on four datasets. Notably, EffoVPR's reranking method outperforms AnyLoc, the current state-of-the-art zero-shot method, on two datasets, even with K=5. However, achieving optimal performance with a low K value depends on the quality of the initial ranking success. In the zero-shot setting, the initial ranking is not highly accurate, so reranking performance generally improves as the K value increases.
>
>  |Dataset|Global|AnyLoc|K=5|K=10|K=15|K=20|K=50|K=100|
> |-------|------|------|---|----|----|----|----|-----|
> |Tokyo24/7|62.2|60.6|**74.0**|**77.8**|**81.0**|**82.5**|**88.3**|**90.8**|
> |MSLS-Val|47.7|68.7|58.5|63.5|64.9|65.7|**68.8**|**70.3**|
> |Pitts30k|78.1|87.7|85.4|87.2|**87.8**|**88.3**|**89.3**|**89.4**|
> |Nordland|33.0|16.1|**41.8**|**46.2**|**48.6**|**50.2**|**54.9**|**57.9**|
>
>  **Q2: ablations in the appendix using EffoVPR-R... confirm...**
>
>  **A2:** Thank you for this comment. While most experiments in the ablation study were conducted using EffoVPR-R, a few were performed at the global level (e.g., feature compactness and number of trainable layers). We have ensured that this distinction is clearly stated in the revised version of the paper.
>
>  **Q3: Re-ranking memory**
>
>  **A3:** Thank you. Here we compare the memory requirements of our reranking method with other reranking methods, together with the associated performance on two datasets.
>
>  |Method|Local Features Dim.|Mem. Footprint (GB)|Tokyo 24/7 R@1|MSLS-Val R@1|
> |------|-------------------|-------------------|--------------|------------|
> |R2Former|500x(128+3)|0.025|88.6|89.7|
> |SelaVPR|61x61x128|0.182|94.0|90.8|
> |EffoVPR-R|649x1024|0.254|98.7|92.8|
>
> We have added this evaluation to the Appendix of the revised paper.

---

> ### Comment · Reviewer_P6DW · 2024-12-01
> **Response to review**
>
> Thank you for the response. It addresses my concerns. I maintain my already positive score.

---

> > ### Author Response · Authors · 2024-12-01
> > **Thank you**
> >
> > We are happy that you found our response satisfactory

---

### Official Review · Reviewer_2roT · 2024-11-03

**Soundness:** 3
**Presentation:** 3
**Contribution:** 2
**Rating:** 6
**Confidence:** 5

**Summary:**

This paper proposes a new method, called EffoVPR, to effectively use vision foundation models for visual place recognition. This method leverages the output class token of the DINOv2 model to generate global features to retrieve candidates. It also uses ViT internal attention matrices to obtain local features for re-ranking these candidates. EffoVPR achieves good zero-shot performance. After training on the SF-XL dataset with the EigenPlaces framework, it outperforms previous SOTA methods on several VPR datasets.

**Strengths:**

1.	The methods proposed in the paper are generally novel.
2.	The experiments provided in the paper are sufficient. The author provides the results of other SOTA methods on datasets such as SF-XL. As far as I know, completing such evaluation experiments requires a lot of time and effort.
3.	The performance of the proposed method is really good. It achieves excellent results even with compact global features.

**Weaknesses:**

1.	The methodological contribution of this paper is a little weak. Its training strategy follows the EigenPlaces work. The global feature is the class token (following a linear layer) of the ViT model. The local matching in the re-ranking process is based on the mutual nearest neighbors searching, and it also already exists. I think the authors should state that the previous SelaVPR work also does not require spatial verification in re-ranking. The biggest contribution of this work seems to be the use of the internal self-attention matrices to yield selected local features, which is meaningful but maybe not enough.
2.	The proposed method is trained on SF-XL, which is a very large dataset and time-consuming to train. If training on other datasets, such as GSV-Cities or even MSLS/Pitts30k. Can it still achieve such good results? If the proposed method cannot extend to other training sets, it is better to have an appropriate explanation in the paper.
3.	The proposed method uses 224×224-pixel images for training, while 504×504-pixel images for evaluation. Using the DINOv2-large model to process such high-resolution images is time-consuming and memory-consuming. Although the paper gives results using low-resolution images on MSLS and Tokyo247, I think it is also necessary to give a comparison of GPU memory usage, and better to provide results on more datasets (e.g., Nordland and SF-XL-Night).

Overall, the strengths of this paper outweigh its weaknesses.

**Questions:**

see weaknesses.

---

> ### Author Response · Authors · 2024-11-22
> **Response to Review**
>
> Thank you for your feedback. We are encouraged that you find our approach “novel” and with “really good performance”.
>
> **Q1a: methodological contribution...**
>
> **A1a:** Our paper highlights two primary contributions:
>
> **1. Leveraging ViT’s Intermediate Layers for VPR Matching:** Current VPR reranking methods typically rely on training modules atop model output tokens. In contrast, traditional general matching algorithms consist of a simpler approach involving keypoint selection and descriptor extraction. Our exploration revealed that the attention maps from ViT’s intermediate layers encapsulate relevant information for this purpose. Specifically, we found that the Q and K facets serve as high-quality keypoint selectors, while the V facet provides robust descriptors. This straightforward yet effective method enhances performance in both zero-shot and fine-tuning scenarios, outperforming current approaches.
>
> **2. Challenging the Need for Complex External Pooling Layers:** Many state-of-the-art methods depend on sophisticated, learnable pooling layers that often result in large global features (e.g., NetVLAD and GeM). We question the necessity of these layers and propose a simple yet highly effective alternative. Our approach not only enhances performance but also significantly improves feature compactness. We believe that this perspective offers valuable insights to the VPR community regarding the role of external pooling layers in modern methods.
>
> These contributions enable us to achieve state-of-the-art results across 20 benchmarks, demonstrating exceptional robustness under diverse conditions—day and night, seasonal changes, and occlusions—while delivering a level of feature compactness that surpasses previous methods by orders of magnitude.
>
> **Q1b: state that SelaVPR works also without geometric verification.**
>
> **A1b:** Thank you. We have added it to the revised version in line 172.
>
> **Q2: training on other datasets**
>
> **A2:** Thank you for this comment. Various VPR methods utilize different datasets for training. Our training approach is based on EigenPlaces, which necessitates a dense coverage of 360 panoramas with orientation (heading) metadata. MSLS, Pitts30k, and GSV-Cities lack the necessary metadata, preventing us from using them for training.
>
> We have added this explanation to the Appendix section A.2.
>
> **Q3: GPU memory usage and results on Nordland and SF-XL-Night**
>
> **A3:** Thank you for the suggestion. Indeed, EffoVPR performs exceptionally well even at lower resolutions. Therefore, it is important to also consider memory footprint and computation time when selecting the input resolution. Below we provide the requested analysis and results. We have extended evaluation on three more datasets, Nordland, SF-XL-night and AmsterTime. Both GPU memory footprint and feature extraction time increase as the resolution rises. Our results remain SoTA across all resolution inputs.
>
> |Resolution|GPU (GB)|Latency (s)|Nordland|SF-XL-night|AmsterTime|
> |----------|--------|-----------|--------|-----------|----------|
> |224px|2.33|0.018|92.7|57.5|65.1|
> |322px|2.38|0.028|94.3|61.2|65.3|
> |504px|2.57|0.067|95.0|61.6|65.5|
>
> We have added these results to the Ablation Study and Table S7 in the Appendix.

---

> > ### Comment · Reviewer_2roT · 2024-11-25
> > **Official Comment by Reviewer 2roT**
> >
> > Thank you for the response. According to the results you provided, when the input image resolution is increased from 224 to 504, the inference time only increases by less than 50%. I have conducted experiments to verify this, and the results are quite different. Please double check it or provide more clarification.

---

> > > ### Author Response · Authors · 2024-11-25
> > >
> > > Thank you! Following your comment, we revisited our runtime computation, and discovered that the initial measurements were taken without GPU synchronization. At that time, the results seemed reasonable due to parallelization capabilities of the GPU. However, after correcting this and ensuring proper GPU synchronization, we have run our test again and validated it also with synthetic data, and obtained the updated runtime in the table.
> > >
> > > For a 504px resolution, the corrected runtime is 0.067 seconds per image, with lower resolutions being notably faster. It is important to clarify that this runtime corresponds to the encoding process. In VPR tasks, the gallery is typically pre-computed (encoded offline), meaning that during retrieval, only the nearest neighbor search and reranking steps contribute to the runtime.
> > >
> > > That being said, please note that EffoVPR maintains state-of-the-art performance on the requested datasets even at lower resolutions.
> > >
> > > We sincerely apologize for this mistake in the rebuttal and appreciate your understanding.

---

### Official Review · Reviewer_ZQCU · 2024-11-05

**Soundness:** 3
**Presentation:** 3
**Contribution:** 2
**Rating:** 6
**Confidence:** 4

**Summary:**

This paper presents an effective approach for visual place recognition (VPR) using a foundation model (DINOv2). The features extracted from the self-attention layers are used as re-ranker for VPR. Experiments on several datasets show the proposed method is robust and effective.

**Strengths:**

1. The proposed method is easy to follow. It leverages the foundation model’s feature representation capability to improve the effectiveness of image retrieval and re-ranking.

2. The proposed method outperforms some existing approaches on several VPR benchmarks.

**Weaknesses:**

1. Since the proposed method mainly leverages the strong representation power of a foundation model (DINOv2), the technical novelty of the proposed method in VPR is somewhat incremental.
2. Certain design choices, such as using the V_l matrix for keypoint descriptors, lack clear motivation and justification, particularly in explaining why such choices are effective.
3. Although the proposed method outperforms several existing methods, the improvement could come solely from using a stronger foundation model feature representation. Therefore, the comparison may not be fair.
4. The computational complexity and memory footprint of the proposed method should be discussed and compared with other existing methods.
5. It would be good to also show some failure cases and provide analysis.

**Questions:**

Please see the weakness section.

---

> ### Author Response · Authors · 2024-11-22
> **Response to Review - Part 1/2**
>
> Thank you for your detailed and insightful feedback and finding our paper "easy to follow" and "improving the effectiveness of image retrieval". We address all your concerns below.
>
> **Q1: Proposed method based on foundation model... technical novelty incremental...**
>
>
> **A1:** Please note that most current SoTA VPR methods build upon strong foundation model (DINOv2) [1][2][3]. We just follow this line of work. However, our contribution extends beyond the use of strong foundation models and lies in two key novel aspects.
>
> 1. Challenging the prevalent practice in foundation-based models of relying on external aggregation modules and questioning their necessity.
>
> 2. Unlocking the encapsulated strength of foundation models, in their intermediate layers.  Here we introduce a novel keypoint extraction and descriptor computation module, based on different K, Q and V self-attention facets extracted from intermediate layers. These keypoints and descriptors demonstrate significant robustness, making them highly effective even in zero-shot setting (surpassing the SoTA method of AnyLoc) and excels in reranking after fine-tuning.
>
> We ultimately achieve state-of-the-art performance across 20 benchmarks. We believe these contributions are significant and far from trivial.
>
> [1] Lu, Feng, et al. "Towards Seamless Adaptation of Pre-trained Models for Visual Place Recognition." In ICLR, 2024.
>
> [2] Lu, Feng, et al. "CricaVPR: Cross-image Correlation-aware Representation Learning for Visual Place Recognition." In CVPR, 2024.
>
> [3] Izquierdo, S., & Civera, J. (2024). Optimal transport aggregation for visual place recognition. In CVPR, 2024.
>
> **Q2: Design choices... lack clear motivation and justification.**
>
> **A2:** Thank you for your comment. We believe that we addressed this concern in our original submission, in the Related Work section (lines 192-193), where [4] and [5] provide key motivation and intuition for our approach. To elaborate further, we draw attention to AnyLoc, which emphasizes that per-pixel features from off-the-shelf foundation models [1] demonstrate remarkable visual and semantic consistency.
>
> Additionally, prior studies have extensively highlighted the robust properties of SSL-trained models, including their semantic and instance-level capabilities [1,2,3]. For instance, [2,4,5] studies investigate various aspects of feature representation, in appearance and semantic properties, while [5] specifically discusses the roles of K, Q, and V facets in such tasks. Building on these insights, we are the first to analyze and reveal their potential for VPR matching.
>
> Our approach leverages K and Q for keypoint detection and utilizes values (V) as descriptors for these keypoints. A detailed analysis of these design choices is provided in the Appendix (Table S3, referenced in line 304, in the original submission).
>
> To emphasize this motivation more clearly, we have slightly revised the introduction section, dedicating a separate paragraph to address this issue.
>
> [1] M. Oquab, T. Darcet et al., “Dinov2: Learning robust visual features without supervision,” arXiv:2304.07193, 2023.
>
> [2] M. Caron, H. Touvron, I. Misra et al., “Emerging properties in selfsupervised vision transformers,” in ICCV, 2021.
>
> [3] N. Park, W. Kim, B. Heo, T. Kim, and S. Yun, “What do selfsupervised vision transformers learn?” in ICLR, 2023
>
> [4] Shir Amir, Yossi Gandelsman, Shai Bagon, and Tali Dekel. Deep ViT features as dense visual descriptors. ECCVW, 2022.
>
> [5] Narek Tumanyan, Omer Bar-Tal, Shai Bagon, and Tali Dekel. Splicing vit features for semantic appearance transfer. In CVPR, 2022.
>
> **Q3: Improving could come from stronger foundation...**
>
> **A3:** Please note that we compare our method with several recent approaches that all utilize DINOv2, including SelaVPR (ICLR 2024), CricaVPR (CVPR 2024) and SALAD (CVPR 2024). Additionally, we incorporated another model formally published after the ICLR deadline – VLAD-BuF (ECCV 2024), which is also based on DINOv2. These comparisons are provided below, and we believe they ensure a fair and consistent evaluation.
>
> |Method|Pitts250k|Tokyo24/7|MSLS-Val|Nordland|AmsterTime|
> |------|---------|---------|--------|--------|----------|
> |VLAD-BuF|95.6|96.5|92.4|78.0|61.4|
> |EffoVPR|**97.0**|**98.7**|**92.8**|**95.0**|**65.5**|

---

> ### Author Response · Authors · 2024-11-22
> **Response to Review Part 2/2**
>
> **Q4: Computational complexity and memory footprint **
>
> **A4:** Thank you for your comment. We provide a breakdown of the memory footprint in the table below, which has been added to the revised version of the paper. The table reports memory usage for 1 million images based on the corresponding feature vector size, along with the equivalent memory footprint for a real-world scenario, using the SF-XL gallery (2.8M images) that covers the city of San Francisco.
>
> Method|Features Dim.|1M images (GB)|SF-XL Gallery (GB)|
> |------|-------------|--------------|------------------|
> |SelaVPR|1024|3.81|10.7|
> |CricaVPR|4096|15.26|42.81|
> |SALAD|8448|31.47|88.3|
> |BoQ|16384|61.04|171.25|
> |EffoVPR-G|1024|3.81|10.7|
> |EffoVPR-G|256|0.98|2.68|
> |EffoVPR-G|128|0.49|1.34|
>
>  We compare our method’s reranking runtime and memory footprint with several other methods as reported in their corresponding papers:
>
>
> |Method|Local Features Dim.|Mem. Footprint (GB)|Latency (s)|Device|Tokyo 24/7 R@1|MSLS-Val R@1|
> |------|-------------------|-------------------|----------|------|-------------|-----------|
> |R2Former|500x(128+3)|0.25|0.202|RTX A5000|88.6|89.7|
> |SelaVPR|61x61x128|0.182|0.085|RTX 3090|94.0|90.8|
> |EffoVPR-R|649x1024|0.254|0.035|A100|98.7|92.8|
>
>
> We have also conducted an evaluation of GPU memory footprint and runtime, analyzing performance across different input sizes. SelaVPR and CricaVPR used 224px resolution and SALAD 322px. EffoVPR-R achieves SoTA performance over all resolutions.
>
> Resolution|GPU (GB)|Latency (s)|Nordland|SF-XL-night|AmsterTime|
> |----------|--------|-----------|--------|-----------|----------|
> |224px|2.33|0.018|92.7|57.5|65.1|
> |322px|2.38|0.028|94.3|61.2|65.3|
> |504px|2.57|0.067|95.0|61.6|65.5||
>
> We have added a special section in Appendix (Section F) presenting details on memory footprint and runtime
>
> **Q5: Failure Cases**
>
> **A5:** Thank you! We have added several failure cases, categorizing and discussing these cases in the Appendix (section D.3) of the revised version.

---

> > ### Comment · Reviewer_ZQCU · 2024-11-26
> > **Post rebuttal**
> >
> > The authors addressed my comments and I have raised my score.

---

> ### Author Response · Authors · 2024-11-28
>
> Thank you for your time and effort in reviewing our paper and rebuttal. We are happy to have successfully addressed your concerns and appreciate the increased score.

---

### Author Response · Authors · 2024-11-22
**Thanking the reviewers**

Dear Reviewers and ACs,

We were happy to see that reviewers found our approach, **“clear and easy to follow”** (ZQCU, P6DW), **“novel”, “Experiments sufficient”, “excellent results even with compact global features”**  (2roT),  **“effective/impressive in zero-shot setting”** (2roT, KXCa),  **“performs significantly better on the more challenging datasets”** and **“training efficient”** (KXCa).

We have addressed the reviewers' concerns in our rebuttal and are open to further discussion. Your input has been instrumental in improving our paper.

In response to the comments, we have uploaded a revised version highlighting changes in blue.

Thank you!

---

### Meta-Review · Area_Chair_oikQ · 2024-12-18

**Metareview:**

This paper introduces a novel method for visual place recognition (VPR) called EffoVPR, which utilizes the DINOv2 foundation model for image feature extraction. EffoVPR is claimed to achieve superior performance on several VPR benchmarks, including the SF-XL dataset, compared to previous state-of-the-art (SOTA) methods. The proposed method is evaluated under different conditions, including zero-shot settings, where it shows promising results without requiring fine-tuning.

All reviewers have given acceptance scores and unanimously agree that the responses during the rebuttal period addressed their concerns. Therefore, I ultimately recommend accepting the paper.

**Additional Comments On Reviewer Discussion:**

All reviewers have given acceptance scores and unanimously agree that the responses during the rebuttal period addressed their concerns. Therefore, I ultimately recommend accepting the paper.

---

### Decision · Program_Chairs · 2025-01-22

Accept (Poster)